# Glutamate dehydrogenase (Gdh2)-dependent alkalization is dispensable for escape from macrophages and virulence of *Candida albicans*

**Fitz Gerald S. Silao**[1], **Kicki Ryman**[1], **Tong Jiang**[2,3], **Meliza Ward**[1], **Nicolas Hansmann**[1], **Chris Molenaar**[1], **Ning-Ning Liu**[4], **Changbin Chen**[2], **Per O. Ljungdahl**[1]*

1 Department of Molecular Biosciences, The Wenner-Gren Institute, Stockholm University, Sweden, 2 The Center for Microbe, Development and Health, Key Laboratory of Molecular Virology and Immunology, Institut Pasteur of Shanghai, Chinese Academy of Sciences, China, 3 University of Chinese Academy of Sciences, China, 4 Center for Single-Cell Omics, School of Public Health, Shanghai Jiao Tong University School of Medicine, Shanghai, P. R. China

* per.ljungdahl@su.se

**Data Availability Statement:** All relevant data are within the manuscript and its Supporting Information files.

## Abstract

*Candida albicans* cells depend on the energy derived from amino acid catabolism to induce and sustain hyphal growth inside phagosomes of engulfing macrophages. The concomitant deamination of amino acids is thought to neutralize the acidic microenvironment of phagosomes, a presumed requisite for survival and initiation of hyphal growth. Here, in contrast to an existing model, we show that mitochondrial-localized $NAD^+$-dependent glutamate dehydrogenase (*GDH2*) catalyzing the deamination of glutamate to α-ketoglutarate, and not the cytosolic urea amidolyase (*DUR1,2*), accounts for the observed alkalization of media when amino acids are the sole sources of carbon and nitrogen. *C. albicans* strains lacking *GDH2* (*gdh2*-/-) are viable and do not extrude ammonia on amino acid-based media. Environmental alkalization does not occur under conditions of high glucose (2%), a finding attributable to glucose-repression of *GDH2* expression and mitochondrial function. Consistently, inhibition of oxidative phosphorylation or mitochondrial translation by antimycin A or chloramphenicol, respectively, prevents alkalization. *GDH2* expression and mitochondrial function are derepressed as glucose levels are lowered from 2% (~110 mM) to 0.2% (~11 mM), or when glycerol is used as primary carbon source. Using time-lapse microscopy, we document that *gdh2*-/- cells survive, filament and escape from primary murine macrophages at rates indistinguishable from wildtype. In intact hosts, such as in fly and murine models of systemic candidiasis, *gdh2*-/- mutants are as virulent as wildtype. Thus, although Gdh2 has a critical role in central nitrogen metabolism, Gdh2-catalyzed deamination of glutamate is surprisingly dispensable for escape from macrophages and virulence. Consistently, using the pH-sensitive dye (pHrodo), we observed no significant difference between wildtype and *gdh2*-/- mutants in phagosomal pH modulation. Following engulfment of fungal cells, the phagosomal compartment is rapidly acidified and hyphal growth initiates and sustained under consistently acidic conditions within phagosomes. Together, our results demonstrate that amino acid-dependent alkalization is not essential for hyphal growth, survival in macrophages and

**Funding:** This work was supported by EU grant MC-ITN-606786 (ImResFun) and grants from the Swedish Research Council VR-2015-04202 and 2019-01547 (POL).The funders had no role in study design, data collection and analysis, decision to publish, or preparation of the manuscript.

**Competing interests:** The authors have declared that no competing interests exist.

hosts. An accurate understanding of the microenvironment within macrophage phagosomes and the metabolic events underlying the survival of phagocytized *C. albicans* cells and their escape are critical to understanding the host-pathogen interactions that ultimately determine the pathogenic outcome.

## Author summary

*Candida albicans* is a commensal component of the human microflora and the most common fungal pathogen. The incidence of candidiasis is low in healthy populations. Consequently, environmental factors, such as interactions with innate immune cells, play critical roles. Macrophages provide the first line of defense and rapidly internalize *C. albicans* cells within specialized intracellular compartments called phagosomes. The microenvironment within phagosomes is dynamic and ill defined, but has a low pH, and contains potent hydrolytic enzymes and oxidative stressors. Despite the inhospitable conditions, phagocytized *C. albicans* cells catabolize amino acids to obtain energy to survive and grow. Here, we have critically examined amino acid catabolism and ammonia extrusion in *C. albicans*, the latter is thought to neutralize the phagosomal pH and induce the switch of morphologies from round "yeast-like" to elongated hyphal cells that can pierce the phagosomal membrane leading to escape from macrophages. We report that Gdh2, which catalyzes the deamination of glutamate to α-ketoglutarate, is responsible for the observed environmental alkalization when *C. albicans* catabolize amino acids *in vitro*. However, the phagosomes formed as macrophages engulf wildtype or *gdh2-/-* cells rapidly become acidified, indicating that Gdh2 has no apparent role in modulating phagosomal pH. Strikingly, and similar to wildtype cells, *gdh2-/-* cells initiate and sustain hyphal growth enabling them to escape from macrophages. Also, Gdh2 is dispensable for virulent growth in systemic models of infection. These results provide new insights into host-pathogen interactions that determine the pathogenic outcome of *C. albicans* infections.

## Introduction

*Candida albicans* is a benign member of mucosal microbiota of most humans. However, in individuals with an impaired immune response, *C. albicans* can cause serious systemic infections associated with high rates of mortality [1,2]. In establishing virulent infections, *C. albicans* cells overcome potential obstacles inherent to the microenvironments in the host. Consistently, the capacity of *C. albicans* to establish a wide spectrum of pathologies is attributed to multiple virulence factors, one of which involves morphological switching from the yeast to filamentous forms (i.e., hyphae and pseudohyphae), reviewed in [3–5]. The ability to switch from yeast to filamentous growth is required for tissue invasion and escape from innate immune cells, such as macrophages, whereas, the yeast form facilitates dissemination via the bloodstream. In addition to escaping from innate immune cells, fungal cells must successfully compete with host cells and even other constituents of the microbiome to take up necessary nutrients for growth [6].

Phagocytes, such as macrophages, are among the first line of host defenses encountered by *C. albicans* (reviewed in [7]). These innate immune cells recognize specific fungal surface antigens via specific plasma membrane-bound receptors [8]. Once recognized, fungal cells are enveloped by membrane protrusions that form the phagosomal compartment. The phagosome

matures by fusing with discrete intracellular organelles, resulting in a compartment with potent hydrolytic enzymes, oxidative stressors and low pH [8–10]. Acidification is important to optimize the activity of the hydrolytic enzymes that target to the phagosome.

*C. albicans* can survive and even escape macrophage engulfment. This is thought to depend on the ability of fungal cells to raise the phagosomal pH via ammonia extrusion. It has been suggested that urea amidolyase (Dur1,2), localized to the cytoplasm, catalyzes the reactions generating the ammonia extruded from cells by the plasma membrane-localized Ato proteins [11,12]. In addition to impairing the activity of pH-sensitive proteolytic enzymes, phagosomal alkalization is thought to initiate and promote hyphal growth [11,13]. Consistent with this notion, *C. albicans* lacking *STP2*, encoding one of the SPS (Ssy1-Ptr3-Ssy5) sensor controlled effector factors governing amino acid permease gene transcription [14], fail to form hyphae and escape macrophages [13]. These observations led to a model that the reduced capacity of *stp2Δ* strains to take up amino acids limits the supply of substrates of Dur1,2 catalyzed deamination reactions, which would result in the reduced capacity to alkalinize the phagosome [12,13].

We have recently shown that mitochondrial proline catabolism is required for hyphal growth and macrophage evasion. The proline catabolic pathway is the primary route of arginine utilization [15] and operates independently of the cytosolic Dur1,2-catalyzed urea-$CO_2$ pathway [15,16]. In contrast to the proposed model [12], we observed that *dur1,2-/-* cells retain the capacity to alkalinize a basal medium containing arginine as sole nitrogen and carbon source [15]. Furthermore, strains carrying *put1-/-* or *put2-/-* mutations exhibit strong growth defects and consequently, are incapable of alkalinizing the same medium, suggesting that alkalization is linked to proline catabolism.

A potential source of ammonia responsible for alkalization is the deamination of glutamate, a metabolic step downstream of Put2. In *Saccharomyces cerevisiae*, the primary source of free ammonia is generated by the mitochondrial-localized $NAD^+$-dependent glutamate dehydrogenase (Gdh2) catalyzed deamination of glutamate to α-ketoglutarate, a reaction that generates NADH and $NH_3$ [17]. Importantly, the reaction is anaplerotic and replenishes the tricarboxylic acid (TCA) cycle with α-ketoglutarate, a key TCA cycle intermediate between isocitrate and succinyl CoA, and an important precursor for amino acid biosynthesis.

Here, we have examined the role of Gdh2 in morphological switching under *in vitro* conditions in filament-inducing media, *in situ* in the phagosome of primary murine macrophages, and in virulence in two model host systems. We show that when *C. albicans* utilize amino acids as sole nitrogen- and carbon-sources they extrude ammonia, which originates from Gdh2-catalyzed deamination of glutamate. In contrast to current understanding regarding the importance of phagosomal alkalization, we report that *C. albicans* strains lacking *GDH2* filament and escape the phagosome of engulfing macrophages at rates indistinguishable to wild-type. Furthermore, we also report that the Gdh2-catalyzed reaction is dispensable for virulence in both fly and murine models of systemic candidiasis.

## Results

### *C. albicans GDH2* is responsible for amino acid-dependent alkalization *in vitro*

Arginine is rapidly converted to proline and then catabolized to glutamate in the mitochondria through the concerted action of two enzymes, proline oxidase (Put1; proline to $\Delta^1$-pyrroline-5-carboxylate or P5C) and P5C dehydrogenase (Put2; P5C to glutamate) (**Fig 1A**). *C. albicans* strains lacking *PUT1* (*put1-/-*) and/or *PUT2* (*put2-/-*) are unable to grow efficiently in minimal medium containing 10 mM of arginine as sole nitrogen and carbon source (YNB+Arg,

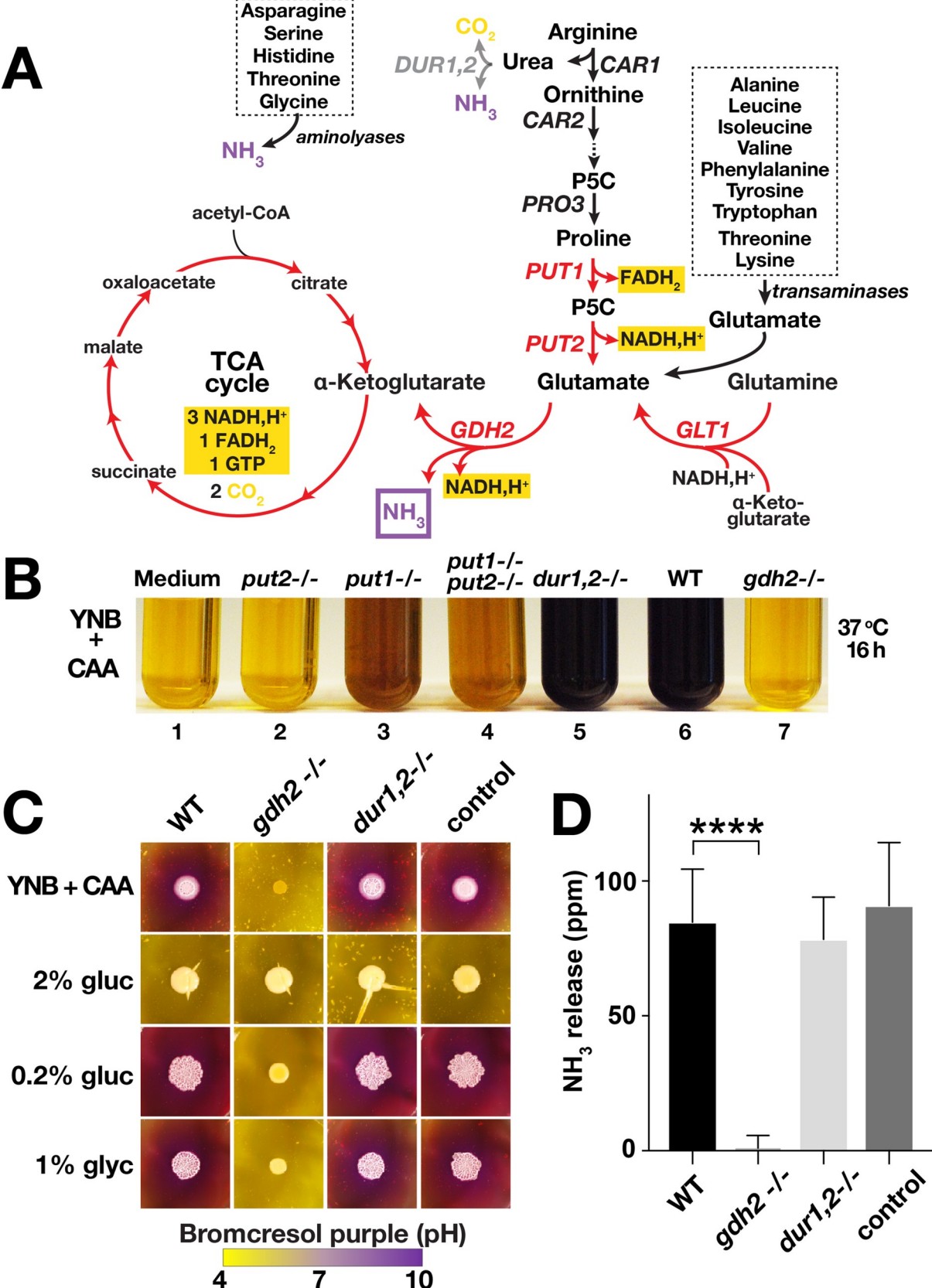

**Fig 1. *C. albicans GDH2* is required for growth using amino acids as sole carbon and nitrogen sources.** (A) Schematic diagram of arginine/ proline catabolism. Arginine is catabolized to proline in the cytoplasm, proline is transported into mitochondria, proline is catabolized to glutamate in two enzymatic reactions, catalyzed by FAD-dependent proline oxidase (*PUT1*) and NAD$^+$-linked $\Delta^1$-pyrroline-5-carboxylate (P5C) dehydrogenase (*PUT2*), respectively. The two central reactions for nitrogen source utilization are catalyzed by NADH-dependent glutamate synthase (*GLT1*) and NAD$^+$-linked glutamate dehydrogenase (*GDH2*). The gene products and metabolic steps marked in red are localized to the mitochondria. (B) YPD grown *put2-/-* (CFG318), *put1-/-* (CFG154), *put1-/- put2-/-* (CFG159), *dur1,2-/-* (CFG246), wildtype (WT, SC5314) and *gdh2-/-* (CFG279) cells were washed, resuspended at an OD$_{600}$ ≈ 0.05 in YNB+CAA containing the pH indicator bromocresol purple, and the cultures were incubated shaking at 37˚C for 16 h. (C) Wildtype (WT, SC5314), *gdh2-/-* (CFG279), *dur1,2-/-* (CFG246) and CRISPR control (CFG182) cells were pre-grown in YPD, washed, resuspended at an OD$_{600}$ ≈ 1, and 5 μl were spotted onto the surface of solid YNB + CAA with bromocresol purple without and with the indicated carbon source. The plates were incubated at 37˚C for 72 h. The images are representative of at least 3 independent experiments. (D) Volatile ammonia released from strains as in (C); the results are the average of at least 3 independent experiments (Ave. ± CI; **** p ≤ 0.0001).

pH = 4.0), and fail to alkalinize this medium [15]. In contrast, cells carrying null alleles of *DUR1,2* (*dur1,2-/-*) grow robustly and alkalinizes the media [15]. To test if the catabolism of amino acids other than arginine and proline can be used as sole carbon source we examined the growth characteristics of the strains in YNB containing 1% casamino acids (CAA), a medium containing high levels of all proteinogenic amino acids (**Fig 1B**). In this media, *dur1,2-/-* cells grew as wildtype and readily alkalinized the media (compare tube 5 with 6). In contrast, *put1-/-* cells exhibited poor growth and weakly alkalinized the medium (tube 3). Cells lacking Put2 activity (*put2-/-*) did not grow appreciably and the culture medium remained acidic (tube 2). Interestingly, a *put1-/- put2-/-* double mutant strain grew better than the single *put2-/-* mutant (compare tube 4 with 2). The severe growth impairment associated with the loss of Put2 is likely due to the accumulation of P5C, which is known to cause mitochondrial dysfunction [18]. These results indicate that the amino acids metabolized via the proline catabolic pathway are preferentially used as carbon sources when mixtures of amino acids are present, e.g., in casamino acid preparations. The catabolism of these non-preferred amino acids contribute only modestly to alkalization, consistent with reports that not all amino acids can serve as carbon sources and contribute to environmental alkalization [12].

The requirement of proline catabolism for growth suggested that the downstream deamination of glutamate to α-ketoglutarate, catalyzed by glutamate dehydrogenase, provided the metabolite responsible for alkalinizing the media. In *S. cerevisiae*, mitochondrial glutamate dehydrogenase (*GDH2*) is the primary source of free ammonia [17]. The *C. albicans* genome has one gene predicted to encode glutamate dehydrogenase (*GDH2*, C2_07900W), and using CRISPR/Cas9 we inactivated both alleles of this gene in SC5314. Due to the dependence of arginine catabolism to proline utilization pathway, we postulated that arginine cannot be efficiently utilized by strain lacking *GDH2* (i.e., *gdh2-/-*). Therefore, we used growth in or alkalization of YNB+Arg medium as a screening criterion to rapidly identify putative *gdh2-/-* clones among Nou$^R$ transformants for PCR-restriction digest verification. In at least two independent transformations, 20–40% of the total colonies (15–25) screened were unable to alkalinize the medium, and all of these colonies were subsequently found to carry mutated and inactive *GDH2* alleles. By contrast, all colonies that grew and alkalinized the media had intact wildtype *GDH2* alleles. Verification and confirmation of two independent *gdh2-/-* clones are shown in **S1A and S1B Fig**. The *gdh2-/-* strains were viable on YPD or YPG (**S1C Fig**), however, they showed both growth and alkalization defects when amino acids were used as sole nitrogen and carbon sources, such as in YNB+Arg (**S1A Fig**) and YNB+CAA media (**Fig 1B and 1C and S5A Fig**). Consistent with what is known for *S. cerevisiae*, the *gdh2-/-* mutant showed a modest growth defect in media containing glutamate or proline as sole nitrogen source (**S1C Fig**). Due to the ease of using CRISPR/Cas9 in creating null mutants, we were able to create additional *gdh2-/-* strains in other genetic backgrounds, e.g., the *cph1Δ/Δ efg1Δ/Δ* [19] that exhibits a highly impaired capacity to form hyphae under most inducing conditions, but remains

competent to alkalinize amino acid-based media (**S1D Fig**). As in SC5314, inactivation of *GDH2* in *cph1Δ/Δ efg1Δ/Δ* (i.e., *cph1Δ/Δ efg1Δ/Δ gdh2-/-*) abolished its capacity to both grow and alkalinize the medium (**S1D Fig**).

To further examine whether the alkalization-deficient phenotype of *gdh2-/-* mutant is due to poor growth or is directly link to its function, we assessed the capacity of the *gdh2-/-* mutant to grow and alkalinize the external growth milieu on solid YNB+CAA (**Fig 1C**). Similar in liquid culture, the absence of an additional carbon source impaired the growth of cells lacking *GDH2* on YNB+CAA solid medium (**Fig 1C**). By contrast, on YNB+CAA supplemented with 2% glucose the *gdh2-/-* strain formed colonies of similar size as WT, indicating that Gdh2 is dispensable for growth in media with high levels of glucose. Consistent with a requirement of a fermentable carbon source, the *gdh2-/-* strain exhibited slightly reduced growth on media with 0.2% glucose, or the non-fermentable carbon source glycerol (1%). Although able to grow in the presence of an added non-fermentable carbon source, the *gdh2-/-* strain failed to alkalinize the media. In contrast, wildtype (WT), *dur1,2-/-* and CRISPR control cells formed colonies of equal size on all media and exhibited identical capacities to alkalinize the media (**Fig 1C**, compare columns 3 and 4 with 1). The medium described here to assess alkalization contains 1% amino acids (CAA) as sole nitrogen and carbon source, which differs from other formulations of YNB containing 0.5% (38 mM) ammonium sulfate [12]. In ammonium sulfate containing medium, we found that the *gdh2-/-* strain also failed to grow and alkalinize the medium. Indeed, the alkalization defect is very tight, even with a high starting cell density ($OD_{600} \approx 5$) and the addition of 1% glycerol to support growth, the culture remained acidic after 24 h at 37˚C, exceeding the strong alkalization defect reported for *stp2Δ/Δ* [12] (**S2 Fig**).

Due to the tight growth and alkalization phenotype in YNB+CAA medium, we were able to directly select for $GDH2^+$ transformants upon the reintroduction of the wildtype allele into the *gdh2-/-* null strains (**S3 Fig**). The *gdh2-/-* strains, derived from SC5314 and *cph1Δ/Δ efg1Δ/Δ* backgrounds, were transformed with a short DNA fragment carrying the wildtype sequence covering the CRISPR-induced mutation. Transformants were plated on solid YNB+CAA containing the pH indicator bromocresol purple (BCP). After 2–3 days colonies appeared and each colony was capable of alkalinizing its immediate vicinity and became purple in color. (**S3A Fig**). As a control, we transformed the cells with an unrelated gene fragment but no growth was observed (**S3A Fig**). The colonies were found to be heterozygous at the *GDH2* locus (*GDH2/gdh2-*) (**S3B Fig**). We also were able to enrich and recover *GDH2/gdh2-* transformants from liquid YNB+CAA (**S3C Fig**). The phenotypes obtained from these reconstituted strains are stable and occur in a manner indistinguishable to starting wildtype strains, indicating the recessive nature of the CRISPR/Cas induced *gdh2-/-* mutations. Together, these observations confirm that the CRISPR/Cas9 induced mutations affected the *GDH2* locus and were the cause of the growth-related phenotypes; Gdh2 activity is required for the alkalization of external growth environment when cells are grown using amino acids as primary nitrogen and carbon source.

## *GDH2* is required for ammonia extrusion

Next, we analyzed whether the alkalization defect of the *gdh2-/-* mutant was due to the lack of ammonia extrusion. Colonies were grown on solid YNB+CAA with 0.2% glucose medium buffered with MOPS (pH = 7.4) and the levels of volatile ammonia produced were measured; the standard acidic growth medium (pH = 4.0) traps ammonia ($NH_3$) as ammonium ($NH_4^+$), decreasing the level of volatile ammonia and thereby interfering with the assay. As shown in **Fig 1D**, the *gdh2-/-* strain did not release measurable ammonia. Consistent with their ability to alkalinize the growth media (**Fig 1C**), wildtype, *dur1,2-/-* and CRISPR control strains released

substantial and indistinguishable levels of ammonia. Together, these results indicate that the reaction catalyzed by Gdh2 generates the ammonia that alkalinizes the growth environment when *C. albicans* uses amino acids as the primary energy source.

## Environmental alkalization originates in the mitochondria

We recently confirmed that mitochondrial activity in *C. albicans* can be repressed by glucose [15], a finding that is consistent with existing transcriptional profiling data [20]. Consequently, the glucose repressible nature of extracellular alkalization in the presence of amino acids could be linked to glucose repressed mitochondrial function. To examine this notion, we first sought to confirm that Gdh2 localizes to mitochondria. Cells (CFG273) expressing the functional *GDH2-GFP* reporter were grown in synthetic glutamate medium with 0.2% glucose ($SED_{0.2\%}$) and YNB+CAA. The GFP fluorescence in cells grown under both conditions clearly localized to the mitochondria as determined by the precise overlapping pattern of fluorescence with the mitochondrial marker MitoTracker Deep Red (MTR) (**Fig 2A**).

To independently assess the role of mitochondrial activity in the alkalization process, we grew the wildtype cells in standard YNB+CAA medium (without glucose), in the presence of Antimycin A, a potent inhibitor of respiratory complex III. No alkalization was observed in the medium even after 24 h of growth (**Fig 2B, upper panel**). Antimycin A clearly impeded the growth of wildtype cells, which phenocopies the *gdh2-/-* growth in YNB+CAA. To ascertain whether the failure to alkalinize the medium was due to inhibiting mitochondrial respiration and not due to cell death, we harvested the cells from antimycin-treated cultures and suspended them in fresh medium; the cells regained their capacity to alkalinize the medium (**Fig 2B, middle panel**). To further test that the inhibitory effect of antimycin on alkalization is specific to mitochondrial function and not an indirect effect of growth inhibition, we performed the same experiment but starting with a high cell density ($OD_{600} \approx 5$). As shown in **Fig 2B** (**lower panel**), untreated cells alkalinized the media in $< 2.5$ h, whereas antimycin-treated cells failed to neutralize the pH of the media even at the lowest antimycin concentration used. We repeated the same experiment using media containing 38 mM ammonium sulfate and 1% glycerol (**S4 Fig**); Antimycin clearly inhibited alkalization even after 4 h, but after 24 h the media became alkaline, suggesting that as cells grew they absorbed the inhibitor, eventually mitochondria regained their respiratory function resulting in alkalization. These results demonstrate that mitochondrial function is essential for environmental alkalization. We also grew the cells in the presence of chloramphenicol, a potent inhibitor that targets mitochondrial translation by reversibly binding to the 50S subunit of the 70S mitochondrial ribosome in yeast [21]. In the presence of this inhibitor and a low starting cell density, a delay in alkalization was observed even if the cells growth is unimpeded due to the addition of 0.2% glucose (**S5A Fig, right panel**). Again, upon extended incubation, as the effective concentration drops due to residual growth, alkalization was virtually indistinguishable after 48 h. Our results indicate that antimycin A is more potent than chloramphenicol. This can be explained since antimycin A acts on existing Complex III, whereas chloramphenicol acts on mitochondrial translation so the effect depends on gradual reduction of existing proteins. Together with our observation that glucose availability influences Gdh2-dependent growth and alkalization (**Fig 1C and S5A Fig**), our findings support the conclusion that alkalization originates from metabolism localized to mitochondria.

## Gdh2 expression is repressed by glucose

To follow up on the observations that glucose negatively affects Gdh2 activity and Gdh2 is a component of mitochondria (**Fig 2A**), we sought to visualize Gdh2 expression in living cells

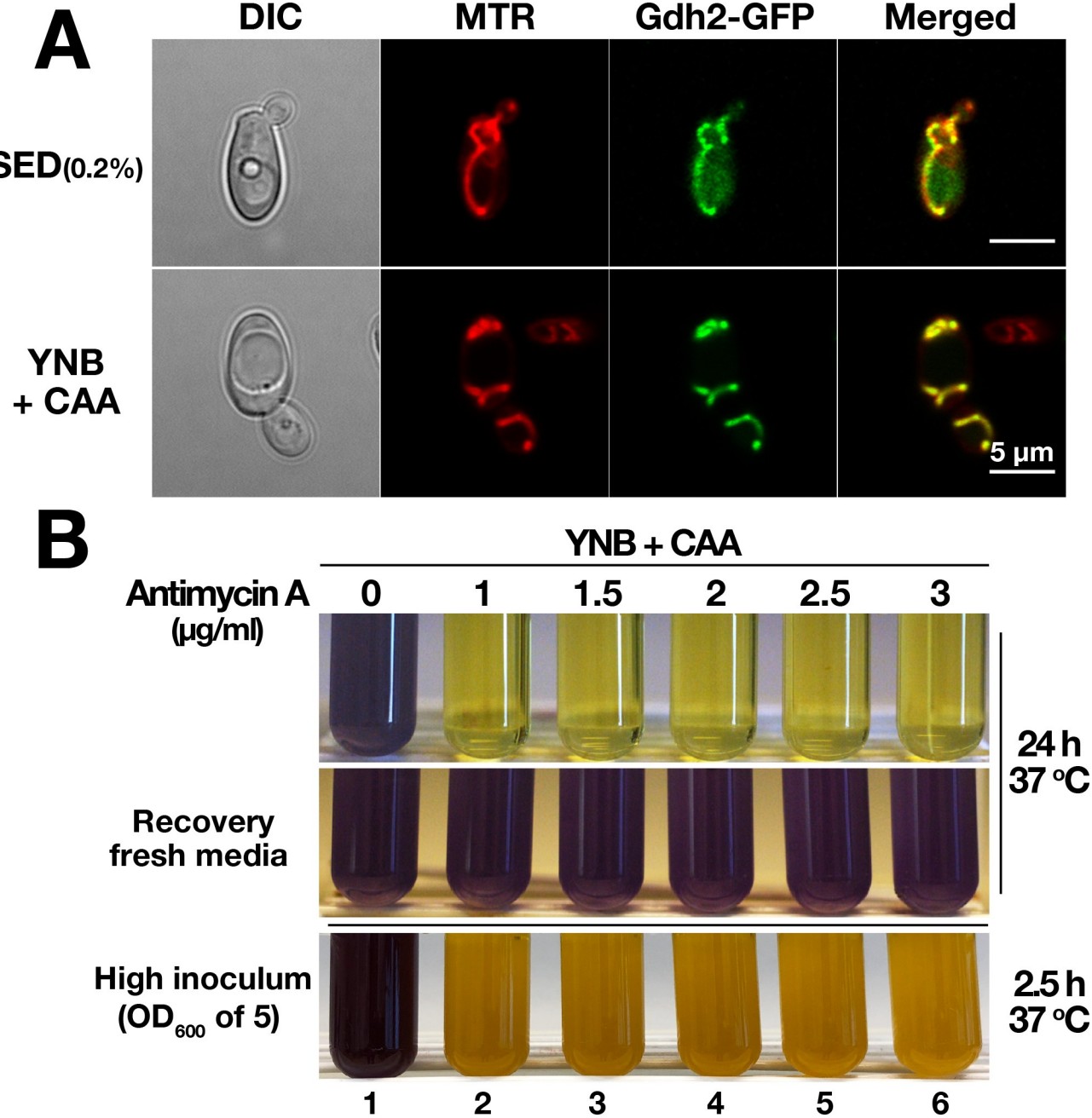

**Fig 2. *C. albicans* Gdh2 localizes to the mitochondria and environmental alkalization requires mitochondrial function.** (A) Gdh2-GFP co-localizes with the mitochondrial marker MitoTracker Deep Red FM (MTR). YPD grown cells expressing *GDH2-GFP* (CFG273) were harvested, washed, grown in SED (0.2% glucose) or YNB + CAA at 37˚C for 24 h, and stained with 200 nM MTR prior to imaging by differential interference contrast (DIC) and confocal fluorescence microscopy; the scale bar = 5 µm. (B) Wildtype cells (SC5314) from overnight YPD cultures were washed and then diluted to either $OD_{600} \approx 0.1$ (top panel) or $\approx 5$ (bottom panel) in liquid YNB+CAA with the indicated concentrations of mitochondrial complex III inhibitor antimycin A. Cultures were grown at 37˚C under constant aeration for 24 h and 2.5 h, respectively, and photographed. To assess viability after Antimycin A treatment, inhibited cells from 24 h old culture (top panel) were harvested, washed, and then resuspended in fresh YNB+CAA media and incubated for 24 h (up to 48 h) at 37˚C (middle panel). Images are representative of at least 3 independent experiments.

when shifted from repressing YPD (2% glucose) to non-repressing YNB+CAA. To do this, we used the same Gdh2-GFP reporter strain described earlier (**Fig 2A**). This enabled us to observe Gdh2 expression at the single cell level over a period of 6 h in cells growing on a thin YNB

+CAA agar slab. The Gdh2-GFP signal was initially weak (t = 0 h), becoming more intense as time progressed and as cells underwent several rounds of cell division (**Fig 3A**).

To relate this observation with the actual alkalization process, we analyzed the levels of Gdh2-GFP in cells grown in liquid culture taken at similar time points. Cells, pre-grown in YPD (2% glucose), were shifted to YNB+CAA and the levels of Gdh2-GFP were assessed by immunoblot analysis. To enable the recovery of adequate amounts of cells for subsequent extract preparation, we increased the starting cell density of the culture (i.e., $OD_{600} \approx 2.0$). As shown in **Fig 3B** (**left panel**) the Gdh2-GFP level in YPD-grown cells was initially low (t = 0 h) but within 2 h the level was greatly enhanced and remained so during the entire 6 h incubation. During the course of growth, the media became successively alkaline, increasing from the starting pH of 4 to 7 (**Fig 3B, right panel**). The finding that Gdh2 expression is induced in cells growing in media rich in amino acids (i.e., YNB+CAA or YPG) indicates that *GDH2* expression in *C. albicans*, in contrast to *S. cerevisiae* [22], is not subject to nitrogen catabolite repression (NCR). NCR is a supra-pathway that prevents the utilization of non-preferred nitrogen source when preferable nitrogen source such as amino acids or ammonium is available [23–25]. This occurs via nuclear exclusion of Gln3 and Gat1 GATA transcription factors from its target promoters.

Next, we examined the expression and stability of Gdh2-GFP in cells shifted from YPD to YPG (**Fig 3C**). The levels of Gdh2-GFP rapidly increased (**lanes 1–3**) and remained high following the addition of glucose (2% final concentration) (**Fig 3C, lanes 5–6**), an observation reminiscent of isocitrate lyase (Icl1), a glyoxylate cycle enzyme that is not subject to catabolite inactivation in *C. albicans* [26,27]. To further investigate the effect of glucose on Gdh2-GFP expression, we shifted YPD grown cells to YNB+CAA in the presence of 0.2% glucose or 1% glycerol, conditions that are not repressive to mitochondrial function (15). As shown in **Fig 3C (upper panels)**, the level of Gdh2-GFP increased substantially, although higher in 1% glycerol than 0.2% glucose. An obvious increase in Gdh2-GFP levels were also observed in cells shifted from YPD to YNB+CAA supplemented with 2% glucose, but relative to the levels in cells shifted to YNB+CAA with 0.2% glucose or 1% glycerol, the Gdh2-GFP levels were low even after 6 h, and it is important to note that the low levels of Gdh2 are insufficient to trigger alkalization (**Fig 3C, lower panel**). We complemented this experiment with microscopy to visualize the expression of Gdh2 (Gdh2-GFP) in wildtype cells (**S6A and S6B Fig**). Consistent with the results from the immunoblot analysis, we observed a significant increase in fluorescence (measured as mean fluorescence intensity, MFI), which localized to mitochondria following a shift from YPD to YNB+CAA with or without 1% glycerol or 0.2% glucose (**S6B Fig**). By contrast, cells grown in 2% glucose exhibited very weak fluorescence compared to 1% or 0.2% glucose, but still significantly higher than YPD-grown cells. The relatively slower rate of Gdh2-GFP fluorescence increase in cells cultured in the presence of 0.2% glucose compared to 1% glycerol is attributed to a more gradual relief from glucose repression from YPD preculture. Our results clearly demonstrate that Gdh2 expression is sensitive to repression by glucose.

## Inactivation of Gdh2 does not impair morphogenesis

Based on current understanding, the ability of *C. albicans* to alkalinize their growth environments is thought to be important for the induction of hyphal growth. Consistent with this notion, *gdh2-/-* cells formed smooth macrocolonies on YNB+CAA (pH = 4.0) with non-repressing 0.2% glucose and 1% glycerol (**Fig 1C, column 2**). To rigorously test the possibility that the inactivation of Gdh2 would negatively affect morphogenesis, we examined growth on Spider and Lee's media, two standard media used to assess filamentation. These media contain

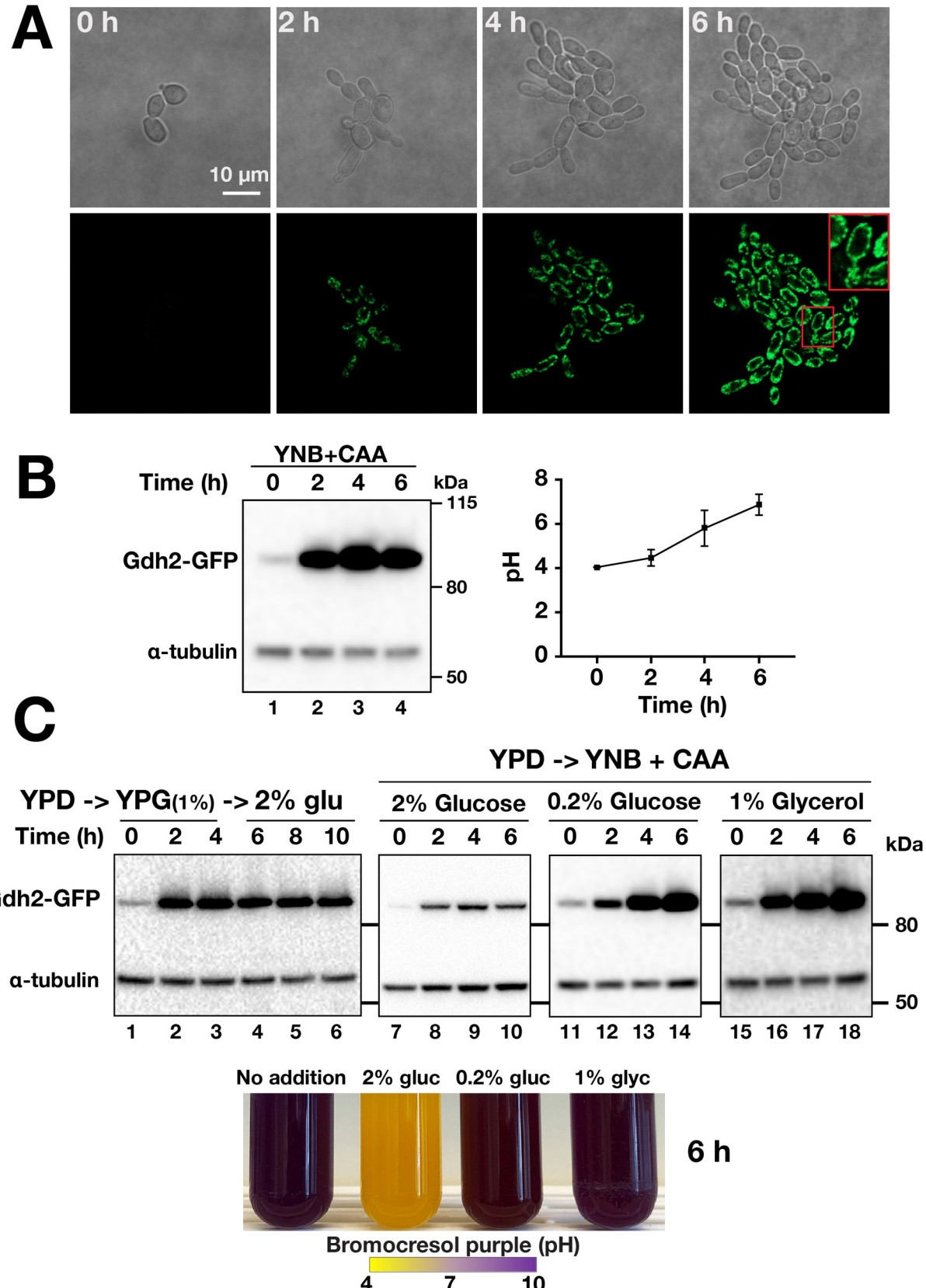

**Fig 3. *GDH2* expression is repressed by glucose.** (A) Live cell imaging of Gdh2-GFP expression in cells (CFG273) shifted from YPD to YNB +CAA. Cells were pre-grown in YPD, transferred to a thin agar slab of YNB+CAA medium and observed by confocal microscopy at 37˚C. Growth and the intensity of mitochondrial-localized GFP signal were monitored every hour for 6 h (images at 2 h intervals are shown). A region in the 6 h micrograph is enlarged (see inset). (B) Gdh2-GFP expression is rapidly induced in cells shifted from YPD to YNB+CAA. Cells were pre-grown in YPD and used to inoculate liquid YNB+CAA ($OD_{600} \approx 2.0$); at the times indicated, the pH was measured (right panel; average of 3 independent experiments) and the levels of Gdh2-GFP expression (left panel) were monitored by immunoblot analysis. (C) Gdh2 expression is carbon source dependent. (upper panels) Cells grown in YPD ($OD_{600} \approx 2.0$) were harvested, transferred to YPG (YP + 1% glycerol; lanes 1–3) and after subsampling at 6 h, 2% glucose was added to cultures (lanes 5–6); YPD grown cells ($OD_{600} \approx 2.0$) were shifted to YNB + CAA with 2% glucose (lanes 7–10), 0.2% glucose (lanes 11–14) or 1% glycerol (lanes 15–18) and grown at 37˚C. Extracts were prepared at the times indicated and the levels of Gdh2-GFP and tubulin (loading control) were assessed by immunoblotting using primary α-GFP and α-tubulin antibodies. (Lower panel) Photograph of tubes after 6 h of growth in YNB+CAA containing BCP without an additional carbon source (No addition), supplemented with 2% glucose, 0.2% glucose or 1% glycerol as indicated.

amino acids, have a neutral pH, and are known to promote filamentous growth of wildtype cells. Similar to wildtype and CRISPR control strains, the macrocolonies formed by the *gdh2-/-* strain were wrinkled and surrounded by an extensive outgrowth of hyphal cells (**Fig 4A**). This indicates that Gdh2 function is dispensable for filamentation, a finding supported by a recent study showing that *gdh2Δ/Δ* strains can undergo morphogenic switching in amino acid-based medium [28].

## Gdh2-GFP expression is rapidly induced upon phagocytosis by macrophages

These results led us to evaluate the capacity of *gdh2-/-* cells to filament within phagosomes of engulfing macrophages. Here too, alkalization is believed to be essential for hyphal growth in the phagosome of engulfing macrophages. First, we determined whether Gdh2 expression is derepressed in phagocytized *C. albicans* cells. The Gdh2-GFP reporter strain (CFG273) was co-cultured with RAW264.7 (RAW) macrophages and phagocytosis events were followed by time-lapse microscopy. The Gdh2-GFP signal intensity increased upon phagocytosis, peaking at around 1 h, and thereafter decreasing during hyphal extension (**Fig 4B; S1 Vid**). We repeated the experiment using primary murine bone marrow-derived macrophage (BMDM) and observed a similar increase in GFP signal. However, due to the inherent green autofluorescence of BMDM, the GFP fluorescence appeared less pronounced (**S2 Vid**). These time-lapse images provide qualitative, yet strong evidence that Gdh2-GFP expression is induced upon engulfment. Attempts to quantify the observed fluorescent signals in the limited number of phagocytized cells were confounded by changes in focal plane and altered mitochondrial dynamics in expanding hyphal cells. To obtain quantitative results regarding the induced expression of Gdh2-GFP within phagosomes, we co-cultured RAW cells with the opsonized reporter strain CFG324 in HBSS for 1 h and analyzed the GFP intensity of both engulfed and external fungal cells. Strain CFG324 is CFG273 (*GDH2/GDH2-GFP*) engineered to constitutively express RFP under the control of the strong *ADH1* promoter ($P_{ADH1}$-*RFP*). The RFP signal was used to locate fungal cells engulfed by macrophages and to normalize the levels of GFP expression. As shown in **Fig 4C**, the Gdh2-GFP signal increased significantly in engulfed fungal cells compared to external cells. The induction of *GDH2* is presumably the reflection of limiting glucose availability and release from glucose repression.

## Gdh2 activity is not required to escape macrophages

We directly compared the ability of wildtype and the *gdh2-/-* mutant cells to survive and escape after being phagocytized by primary BMDM (**Fig 5**). To test this notion, we performed colony forming unit (CFU) assays to quantify the survival of phagocytized cells and time lapse microscopy (TLM) to follow spatio-temporal dynamics of hyphal formation in the phagosomes. The

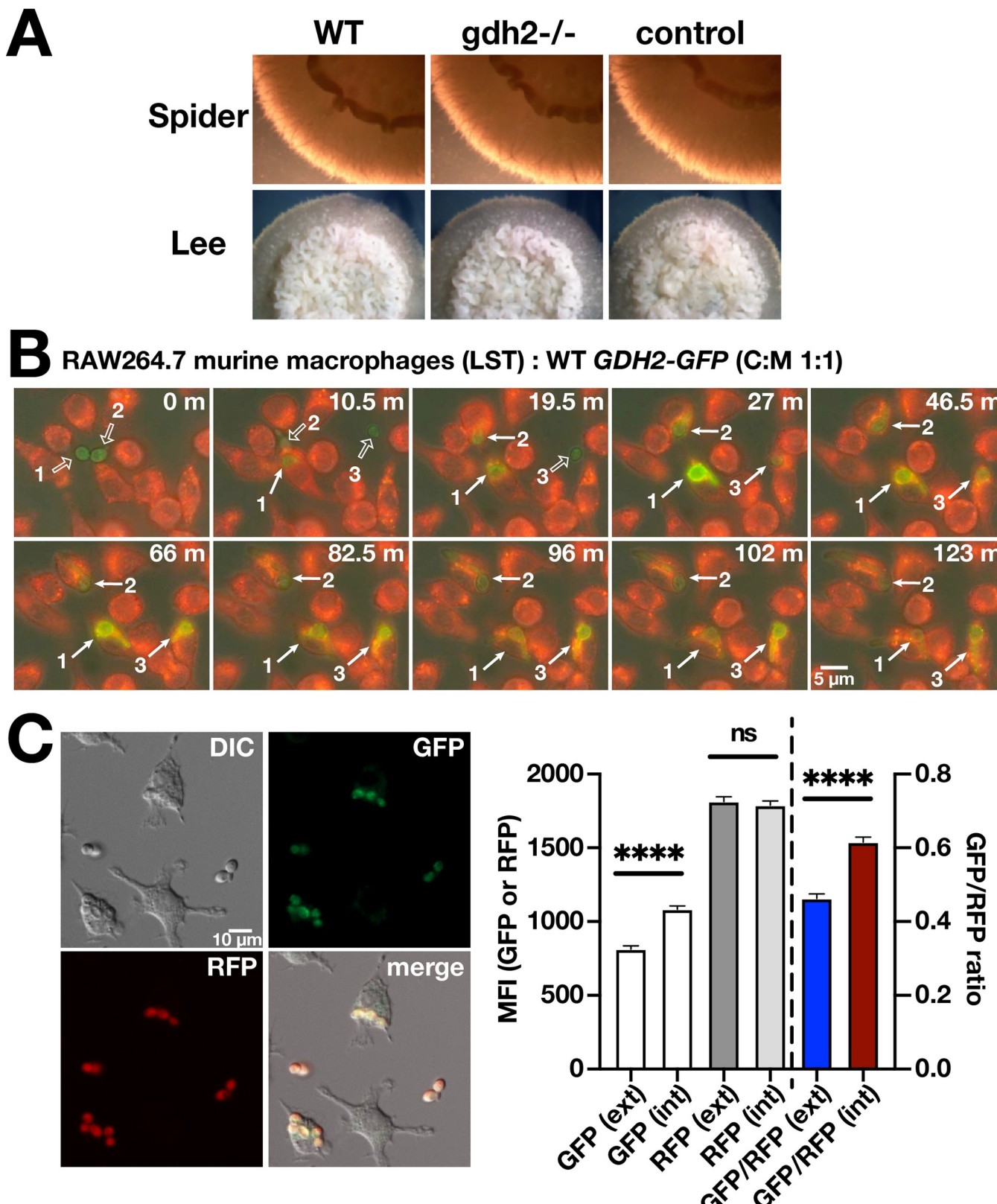

**Fig 4. Gdh2 is dispensable for filamentous growth on solid media and is induced in *C. albicans* cells phagocytized by murine macrophages.** (A) Wildtype (WT, SC5314), *gdh2-/-* (CFG279), and CRISPR control (CFG182) strains, pre-grown in YPD, were washed, resuspended at an $OD_{600} \approx 1$ in water, and 5 µl

aliquots were spotted on solid Spider and Lee's media. Representative colonies were photographed 5 days after incubation at 37°C. CFG273 cells were co-cultured in $CO_2$-independent medium with (B) RAW264.7 macrophages, pre-stained with Lysotracker Red (LST) at MOI of 1:1 (C:M). The co-cultures were followed by live cell imaging. Micrographs were taken at the times indicated (S1 Vid). In each series, three CFG273 cells are marked prior to (open arrows) and after (closed arrows) being phagocytized. (C) Gdh2 expression in phagocytized fungal cells (CFG324) expressing Gdh2-GFP and $P_{ADH1}$-RFP). RAW264.7 macrophages were co-cultured with opsonized CFG324 for 1 h in HBSS prior to measuring GFP and RFP intensities in non- (ext) and phagocytized (int) fungal cells. Results from 3 biological replicates ($\geq$ 100 ext and int cells/replicate) are shown (Ave. ± CI; **** $p \leq 0.0001$ by Student $t$-test).

results show that Gdh2-mediated alkalization is not essential for survival in BMDM (**Fig 5A, right panel**) as there was no significant difference between the two strains. To observe the behavior of each strain following engulfment, we used a wildtype strain constitutively expressing GFP (*ADH1/P_{ADH1}*-GFP) (23) and constructed a *gdh2*-/- mutant strain (CFG275) constitutively expressing RFP (*gdh2*-/- *ADH1/P_{ADH1}*-RFP). Fungal cells were first incubated with BMDM at MOI of 3:1 in HBSS for 30 min, the co-cultures were washed to remove non-phagocytized fungal cells, and fresh HBSS was added and the co-cultures were examined by time lapse microscopy. As shown in **Fig 5A** (**S3 Vid** and **S4 Vid**), both strains exhibited similar characteristics with obvious filamentous growth. For competition assay (see Methodology), equal numbers of WT and *gdh2*-/- cells were mixed (green:red; 1:1), and then added to macrophage at the same MOI of 3:1. The ratio of WT to *gdh2*-/- before and after co-culture (2 h) was assessed via growth (and alkalization) assessment of individual colonies recovered from plating on YPD in YNB+CAA medium in a 96 well format. As shown in **Fig 5B** (**right**), there was no significant difference between the ratio of two strains before and after co-culture. We also monitored the behavior of these strains during co-culture via TLM and found no difference between the two as both managed to escape the BMDM (**Fig 5B, left; S5 Vid**). Together, the data indicate that Gdh2-catalyzed ammonia extrusion and environmental alkalization is not a requisite for initiation of hyphal formation, growth and survival in the phagosome of primary BMDM.

## Hyphal growth of *C. albicans* cells initiates and is sustained within the acidic microenvironment of macrophage phagosomes

We assessed the capacity of wildtype and *gdh2*-/- cells to modulate phagosomal pH using the pH-sensitive dye pHrodo [29–32]. The fluorescence intensity of pHrodo increases significantly as the surrounding pH becomes acidic. Wildtype and *gdh2*-/- cells were individually conjugated with pHrodo in $NaHCO_3$ buffer without the extra methanol washing steps recommended by the manufacturer (see Methodology); we found that the methanol treatment killed the fungal cells. Instead, we performed a series of PBS washing steps to remove excess dye and the fungal cells were opsonized. This labeling protocol did not affect cell viability (**S7A Fig**). To assess the pH dependency of pHrodo fluorescence intensity we placed labeled wildtype and *gdh2*-/- cells in buffers of different pH and measured the cell-associated fluorescence intensities by microscopy and fluorimetry (**S7B Fig**). Fluorescence emission scans of pHrodo labeled wildtype cells exhibited a single peak of pH-dependent fluorescence with the expected emission wavelength of 585 nm; the intensity of the fluorescence was inversely related to pH (**S7C Fig**). No fluorescence was observed in unlabeled cells.

To monitor phagosome acidification, we determined the mean fluorescence intensity (MFI) of phagosomes containing pHrodo labeled fungal cells every 30 min for 4 h. Co-cultures and imaging were performed in isotonic HBSS buffer (pH = 7.4). Under these conditions, in contrast to standard culture media, hyphal formation of non-phagocytized fungal cells is delayed. Importantly, we were able to perform time lapse microscopy for > 12 h without obvious change in overall macrophage health (> 90% viability). The MFI of phagosomes containing wildtype or *gdh2*-/- cells increased following engulfment with the highest intensity recorded after 4 h (**Fig 6A and S6 Vid to S12 Vid**). Based on the pH calibration data (**S7B**

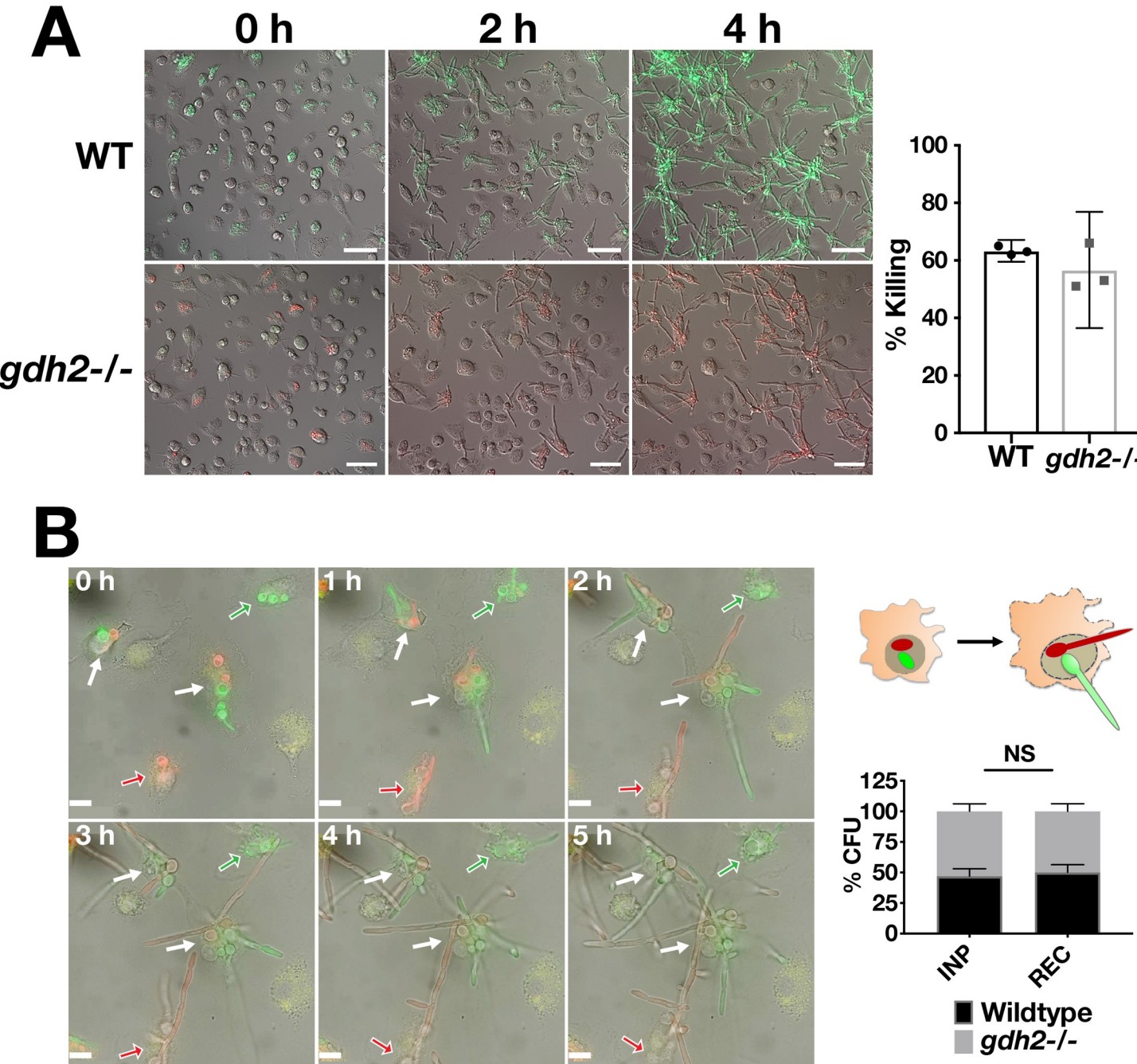

**Fig 5. Competition assay to compare wildtype and *gdh2-/-* filamentation and survival upon phagocytosis by primary BMDM.** (A) (left panels) Wildtype (WT; $P_{ADH1}$-*GFP*; SCADH1G4A) or *gdh2-/-* ($P_{ADH1}$-*RFP*, CFG275) cells was co-cultured with primary BMDM (MOI of 3:1; C:M) for 30 min in HBSS. Non-phagocytized fungal cells were removed by washing and the co-cultures were monitored by live cell imaging for 4 h (**SV3 and SV4 Vids**). (right panel) Candidacidal activity of BMDM. Untagged WT (PLC005) and *gdh2-/-* (CFG279) strains were co-cultured for 2 h with BMDM at MOI of 3:1 (without washing) and then CFUs recovered and compared to the CFUs in the starting inoculum. No significant difference in the co-culture survival between the strains using Student *t*-test; data presented are average of 4 biological replicates. (B) Competition assay. The same strains as in (A) were mixed 1:1, co-cultured for 30 min in HBSS, washed extensively to remove external cells, and the interactions were followed by TLM for 5 h (S5 Vid). Solid arrows indicate macrophages with phagosomes containing both WT and *gdh2-/-* cells; open arrows indicate macrophages with phagosomes containing either WT (green) or *gdh2-/-* (red) cells. The observed growth of WT and *gdh2-/-* cells within a single macrophage is schematically illustrated (right upper panel). For the WT to *gdh2-/-* ratio determination following co-culture, mutants were mixed 1:1 and then the mixed cells were used to infect BMDM at MOI of 3:1. After an initial macrophage lysing step, the ratio of WT:*gdh2-/-* was analyzed by plating serially diluted cells on YPD and then testing at least 50 independent colonies for their ability to alkalinize YNB+CAA medium (pH = 4). The ratio of recovered cells (after 2 h) was compared to the input ratio. No significant difference in the WT:*gdh2-/-* ratio between input and recovered using Student *t*-test; data were obtained from six (6) biological replicates.

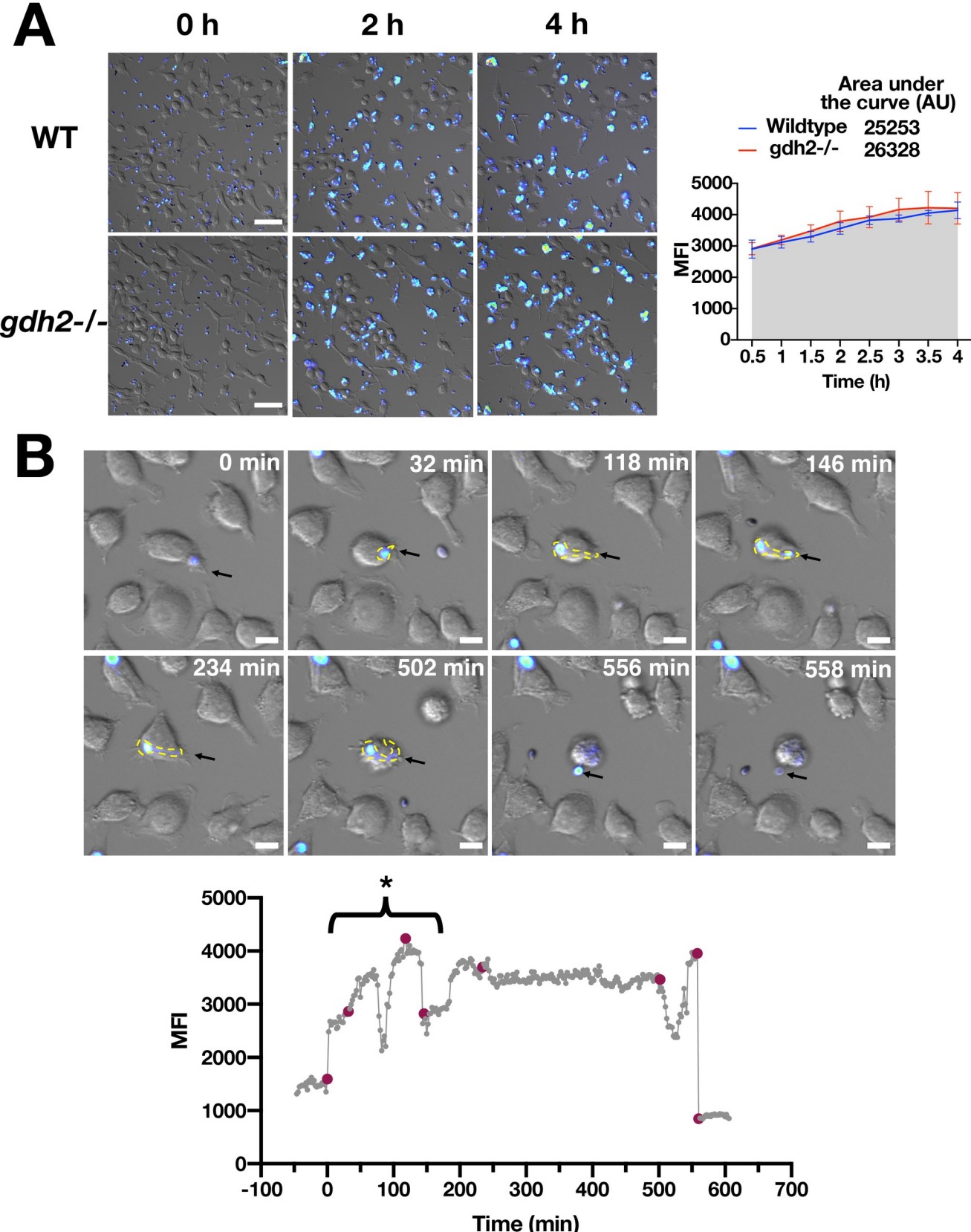

**Fig 6. Phagosomal acidification following phagocytosis.** (A) Wildtype (PLC005) and *gdh2-/-* (CFG279) cells were labeled with pHrodo, and co-cultured with RAW macrophages in HBSS at MOI of 3:1 (C:M); images were obtained at 30 min intervals for 4 h. Representative images and quantification of phagosome intensities for both strains. (B) (upper panels) Time lapse images showing the pH changes encountered by a single fungal cell phagocytosed by a RAW macrophage. Time = 0 min is arbitrarily set when a single *C. albicans* (*gdh2-/-*; CFG279) cell is about to be phagocytized by the macrophage. Black arrow indicates the macrophage containing the fungal cell followed for quantification of fluorescence intensities (lower panel). The time points with maroon color correspond to the time images in the upper panels were taken. The asterisk (*) and bracket indicate a period of time with apparent rapid changes in fluorescence intensity, the consequence of phagosomes moving in and out of focus (S12 Vid).

**Fig**), the phagosomal compartment reached an acidic pH of between 4 and 5. We tested whether increasing the number of target cells in the phagosomes (i.e., high MOI of 6:1 (C:M)) would negatively affect the acidification rate, however, phagosomes containing > 3 fungal cells exhibited intense fluorescence (**S10 Vid** and **S11 Vid**). In control experiments we observed a significant reduction in the MFI 5 min after the addition of ionophores monensin and nigericin (10 μM each) (**S8 Fig**). Monensin and nigericin act as $H^+$/cation exchangers dissipating proton gradients. Thus, the increased fluorescence intensity associated with phagocytized fungal cells is the *bona fide* consequence of a low phagosomal pH.

Consistent with a recent report [33], we observed that hyphal growth initiated in phagosomes when the pH was clearly acidic (**S8 Vid** and **S9 Vid**). We followed the fate of a single cell (*gdh2-/-*), from its initial engulfment to its eventual escape from a macrophage, and monitored the phagosomal pH (**Fig 6B**; **S12 Vid**). Upon engulfment, the fluorescence intensity rapidly increased indicating that the fungal cells were exposed to an acidic environment. The fluorescence intensity remained high for several hours, during which hyphal growth was induced, eventually leading macrophage lysis. Upon becoming re-exposed to the culture media, the pHrodo fluorescence intensity of the escaped fungal cell diminished, reflecting the neutral pH of the medium.

## Gdh2 activity is dispensable for virulence in intact host

Next, we examined the role of Gdh2-dependent alkalization in the capacity of *C. albicans* to successfully infect an intact living host. We used an improved *Drosophila melanogaster* infection model with the *Bom^Δ55C* flies that lack 10 Bomanin genes on chromosome 2 encoding secreted peptides with antimicrobial property [34]. As shown in the survival curve, the *gdh2-/-* mutant remained competent to infect *Bom^Δ55C* flies similar to wildtype control (**Fig 7A**). The data indicate that Gdh2-dependent alkalization is not required for virulence in a *Drosophila* infection model.

To further assess the importance of Gdh2 in virulence, we used a tail vein infection model with C57BL/6 mice as hosts. Two groups of mice (n = 10) were challenged with $3 \times 10^5$ wildtype or *gdh2-/-* cells and survival was monitored for a period of up to 8 days. Similar to the fly model, we observed that the loss of Gdh2 activity did not attenuate virulence (**Fig 7B**); the *gdh2-/-* mutant exhibited survival indistinguishable from wildtype. Consistently, the fungal burden of *gdh2-/* cells in the brain, kidney and spleen of infected mice 3 days post-infection did not significantly differ to mice infected with wildtype (**Fig 7C**). Next we performed a competition assay; equal numbers of wildtype and *gdh2-/-* cells were intravenously injected in mice and the ratio (R) of wildtype to *gdh2-/-* cells recovered from kidneys 3 days post-infection was determined. Consistent to our findings in mice individually infected with each strain, the ratio of recovered cells did not significantly differ to that of the inoculum ratio (I) (**Fig 7C**). Together, our results indicate that Gdh2 is not required for virulence, and that the loss of Gdh2 activity does not create a selective disadvantage or significantly impair growth in infected model host systems.

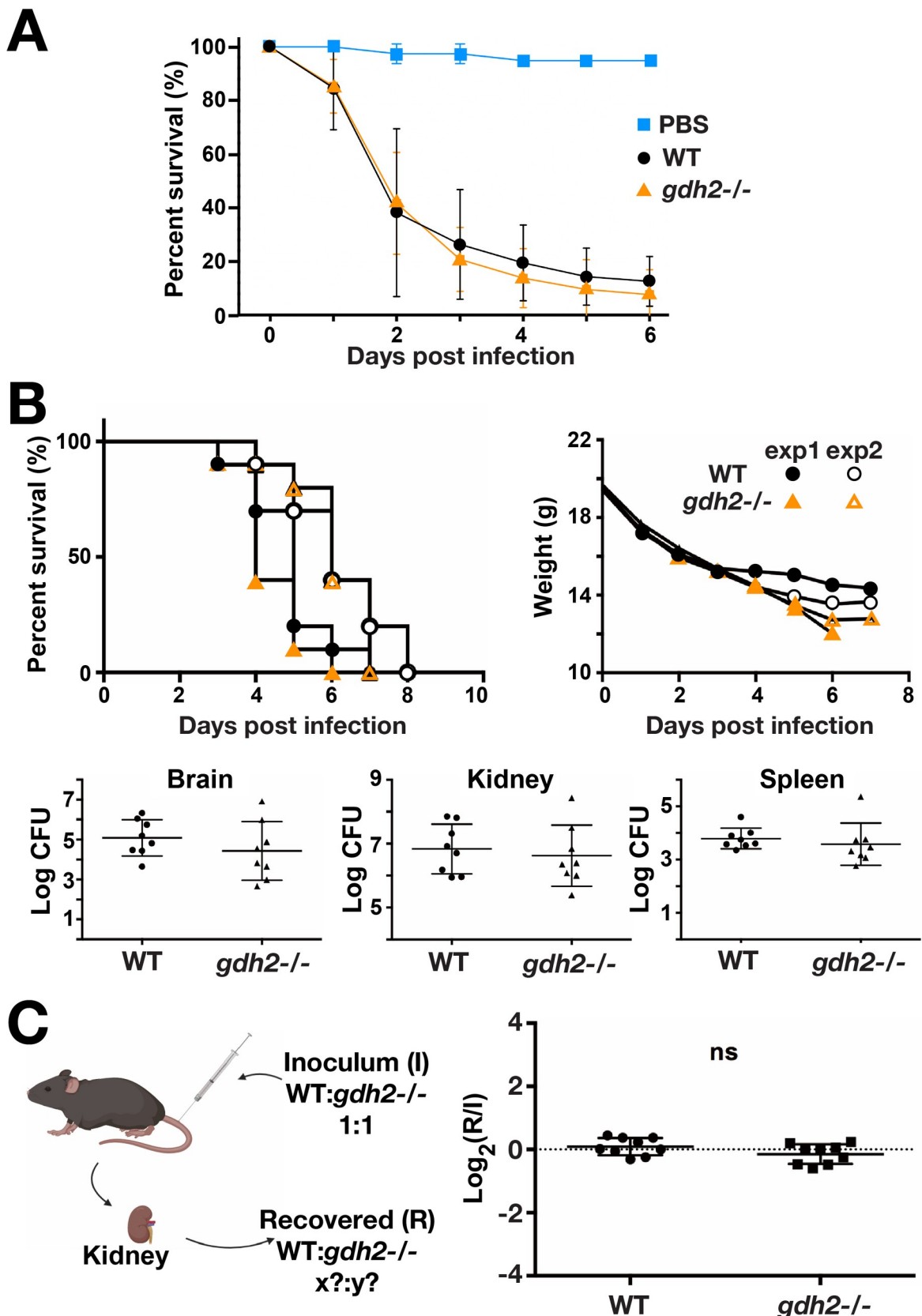

**Fig 7. Virulence of wildtype and *gdh2-/- C. albicans* in *Drosophila* and murine systemic infection models.** (A) *D. melanogaster* *Bom*[Δ55C] flies were infected with wildtype (SC5314) or *gdh2-/-* (CFG279) cells as indicated, and the survival of flies was followed for six days. Each curve represents the average of a minimum of three independent infection experiments (20 flies/strain) performed on different days. (B) Groups of C57BL/6 mice (n = 10) were infected via the lateral tail vein with $3 \times 10^5$ CFU of *C. albicans* wildtype or *gdh2-/-* cells (upper panels) and survival (left) and weight loss (right) was monitored at the timepoints indicated. Survival curves from two independent experiments were statistically analyzed by the Kaplan-Meier method (a log-rank test, GraphPad Prism), no significant difference. The fungal burden (lower panels) in brain (left), kidney (middle), and spleen (right) extracted from mice 3 days post infection. Each symbol represents a sample from an individual mouse and results were compared by Student *t*-test, no significant difference. (C) Competition assay; mice were infected via the tail vein with an inoculum (I) comprised of an equal number of wildtype (SC5314) and *gdh2-/-* (CFG279), 1:1. At 3 days post infection, the abundance and genotype of fungal cells recovered from kidneys was quantitated and the ratio of wildtype:*gdh2-/-* recovered (R) was determined. The significance of the $\log_2(R/I)$ values was assessed using an unpaired Student *t*-test, no significant difference (ns).

## Discussion

In this report we define the precise metabolic step that endows *C. albicans* with the capacity to increase the extracellular pH by ammonia extrusion when cells are grown in the presence of amino acids. We have previously shown that hyphal development of *C. albicans* cells engulfed by macrophages is dependent on proline catabolism in the mitochondria [15]. *In vitro*, strains unable to catabolize proline do not grow and alkalinize media containing arginine as sole nitrogen and carbon source. This led us to test the role of Gdh2 that catalyzes the deamination of glutamate to α-ketoglutarate and functions in the mitochondria as a downstream component of the proline utilization pathway. Our findings indicate that Gdh2 is indeed essential for the observed alkalization of the extracellular milieu when *C. albicans* cells are catabolizing amino acids as nitrogen and energy sources. This result is particularly striking in that the alkalization defect is total even when a very high starting cell density ($OD_{600} \approx 5$) is used. Thus, the defect is more severe than that observed with strains lacking Stp2, the SPS-sensor regulated transcription factor required for efficient amino acid uptake [14]. Strains carrying *stp2Δ/Δ* null mutation have been used extensively due to their decreased ability to increase the pH when grown in the presence of amino acids [12,13,35,36].

Similar to *S. cerevisiae*, the Gdh2-catalyzed reaction is the primary source of ammonia extruded by *C. albicans*. This conclusion is based on the following key observations: 1) mutant strains lacking *GDH2* (*gdh2-/-*) are unable to alkalinize media under non-repressing glucose conditions with amino acids as carbon and nitrogen source (i.e., YNB+CAA) (**Fig 1C** and **S5A and S5B Fig**); 2) ammonia is not extruded by *gdh2-/-* mutants (**Fig 1D**); and 3) the Gdh2-catalyzed reaction occurs in the mitochondria, and consistently, is inhibited by inhibitors of mitochondrial respiration and subject to glucose repression (**Figs 2, 3, S5A Fig** and **S6 Fig**).

Unexpectedly, although Gdh2 is a key enzyme of central nitrogen metabolism, *gdh2-/-* mutants merely exhibit a modest growth defect on synthetic glucose or glycerol media containing glutamate as sole nitrogen source (**S1C Fig**). Consistent with proline being catabolized to glutamate in a linear pathway mediated by Put1 and Put2, a similar modest growth defect of *gdh2-/-* was observed when proline was the sole source of nitrogen (**S1C Fig**). However, when glucose or glycerol is removed from the media (i.e., YNB+CAA), and amino acids serve as both carbon and nitrogen sources, the *gdh2-/-* mutant demonstrated a striking growth defect (**Fig 1B and 1C** and **S5A Fig**). Under these conditions, the impaired growth of cells lacking Gdh2 is likely the consequence of diminished levels of α-ketoglutarate. In the absence of Gdh2, cells must rely entirely on the TCA cycle to generate α-ketoglutarate required to support *de novo* biosynthetic needs, e.g., amino acid biosynthesis. In the absence of glucose or glycerol, the supply of acetyl-CoA derived from pyruvate becomes limiting, stalling the TCA cycle, and consequently, the growth of *gdh2-/-* cells.

The capacity of glucose to repress mitochondrial activity [15] and Gdh2 expression (**Fig 3C** and **S6 Fig**) provides the explanation for alkalization being observed only when glucose

becomes limiting or replaced by glycerol. The induced levels of Gdh2 expression observed upon phagocytosis is consistent with the apparent low levels of glucose in phagosomes [37]. The induction of Gdh2 expression correlates well with our previous finding that Put2 levels are also induced upon phagocytosis [15], which requires proline binding to the transcription factor Put3 [15,38]. The presence of proline in phagosomes and its metabolism in the mitochondria will generate glutamate. Outside of a phagosome context, amino acids present in YNB+CAA, such as glutamine, alanine, and aspartate [24], can be converted directly to glutamate bypassing the requirement for the proline catabolic pathway. However, the fact that *put1*-/- strains exhibit reduced capacities compared to wildtype to alkalinize YNB+CAA indicates that in the absence of glucose, proline functions as a preferred carbon source, which perhaps is due to the consequence of its catabolism being efficiently coupled to the generation of ATP [15].

An unanswered question is how ammonia generated in the mitochondria is extruded to the external environment. It is possible that ammonia ($NH_3$), known to be membrane permeable, diffuses across the inner mitochondrial membrane, moving towards the more acidic inner membrane space where it likely becomes protonated to ammonium. Ammonium then moves to the cytosol. Although the dissociation of ammonium to ammonia is not favored at the pH of the cytosol (pH ~7), the small amount of ammonia that forms can rapidly diffuse across the PM out of cells as long as the external environment is acidic. Hence, the ability of the ammonia generated by Gdh2 to alkalinize the growth environment is likely the consequence of Pma1 activity, the major proton pumping ATPase in the PM. Alternatively, and according to several reports, putative ammonia transport proteins, the Ato family of plasma membrane proteins, are thought to facilitate ammonia export in *C. albicans* [11]. Supporting this notion, the deletion of *ATO5* significantly delays alkalization. The requirement for the Ato proteins suggests that the species traversing the plasma membrane from within the cell is either charged or polar, thus, it is likely, as previously suggested in yeast [39,40], that the transported species is ammonium ($NH_4^+$) and coupled to $H^+$ import. Since cytoplasmic pH is tightly regulated, the conundrum persists as to how extruding ammonium can facilitate steady-state alkalization. The underlying mechanism of how Ato proteins may contribute to alkalization needs to be defined and placed in context to the fact that ammonia can readily diffuse through membranes, and in so doing, is expected to move directionally towards acidic environments where it has an alkalinizing effect.

Strikingly and contrary to our expectations, *GDH2* is dispensable for the induction of hyphal growth and escape of *C. albicans* from macrophages, and for virulence in intact hosts. This contrasts to what we observed in strains lacking *PUT1* and/or *PUT2*, which show a phagosome-specific defect in hyphal growth, a defect that was traced to decreased ATP production [15]. On amino acid-rich Spider medium containing 1% mannitol as primary carbon source, both *put1*-/- and *put2*-/- mutants exhibit noticeable defects on formation of invasive filaments. In contrast, *gdh2*-/- mutants do not show a filamentation defect. Together, these observations suggest that when glucose is limiting, the energy obtained by the catabolism of proline to glutamate suffices to induce and support hyphal growth; the additional energy derived from the NADH generated by the Gdh2-catalyzed deamination of glutamate is not required.

In contrast to previous suggestions [12], we show that *DUR1,2* activity does not significantly contribute to alkalization when cells are grown in the presence of metabolizable amino acids. This is consistent with *DUR1,2* being under tight regulatory control by NCR, and thus, is not expressed under such conditions. Standard growth media (i.e., YPD or SD) used for *C. albicans* propagation and mammalian cell culture media (i.e., DMEM or RPMI) used for co-culturing fungal cells with macrophages, contain high levels of amino acids, and hence represent growth conditions that repress *DUR1,2* expression [25]. The initiation of hyphal formation

following phagocytosis is very rapid, occurring as early as 15–20 min following phagocytosis, which suggests that the signaling cascades initiating morphogenesis are activated early when *DUR1,2* expression is repressed.

We considered two possibilities that could underlie the fact that Gdh2 is dispensable for hyphal growth in and escape from the phagosomes of engulfing macrophages. Either alkalization of the phagosomal compartment is not a requisite for hyphal growth, or that the phagosomal compartment becomes alkaline independent of Gdh2, presumably the consequence of other metabolic processes. To distinguish between these two possibilities, we monitored phagosomal pH using pHrodo labeled *C. albicans* cells and live cell imaging. Consistent with a recent report [33], we observed that hyphal growth initiated in phagosomes when the pH was clearly acidic (**S8 Vid** **and** **S9** **Vid**). We followed the fate of a single pHrodo labeled cell (*gdh2-/-*), from the its initial engulfment to its escape from a macrophage and monitored the fluorescence intensity (**Fig 6B**; **S12 Vid**). Upon engulfment, the fluorescence intensity rapidly increased indicating that the fungal cells were exposed to an acidic environment. The fluorescence intensity remained high for several hours, during which hyphal growth was induced, eventually leading to the macrophage lysis. Upon lysis, the fungal cell was re-exposed to the neutral pH of the culture media and the fluorescence intensity diminished. These observations clearly show that alkalization is not a requisite to induce or maintain hyphal growth of phagocytized *C. albicans* cells.

Our observations that the phagosomal pH remained acidic even when wildtype *C. albicans* cells were examined requires comment since it is not consistent with the current consensus in the field. It has been proposed that phagosomal pH increases as a consequence of rapid hyphal expansion, physically affecting the integrity of the phagosome membrane [33]. The apparent differences could merely be technical; previous measurements of phagosomal pH have relied on dual wavelength ratiometric imaging, whereas our data was obtained using the pH sensitive indicator pHrodo. To our knowledge, this is the first time pHrodo is used to monitor phagosomal pH in macrophages engulfing live *C. albicans* cells. Also, our co-cultures were performed using HBSS, a medium that we believe provides more optimal conditions to observe phagosome-specific events; standard cell culture media readily induces hyphal growth of non-phagocytized cells, potentially obfuscating observation. Interestingly, hyphae in the phagosomes of macrophage in co-cultures carried out in HBSS are relatively short compared to the same time points performed in cell culture media. This raises the question as to how fungal cells within a membrane bounded organelle, such as the phagosome, are affected by the external medium. Our control experiments confirm that the phagocytized fungal cells are located within acidified compartments as the fluorescence intensity dropped following ionophore treatment (**S8 Fig**). The results clearly show that phagosomal alkalization is unlikely to be a defining event initiating hyphal growth of *C. albicans* in phagosomes of engulfing macrophages.

Beyond the realm of the phagosome, our work also suggests that environmental alkalization via ammonia extrusion, a mechanism that is thought to facilitate virulence of fungal pathogens [41,42], is dispensable for pathogenesis of *C. albicans* (**Fig 7**). In contrast to our findings, current understanding suggests that phagosomal alkalization via ammonia extrusion is critical to *C. albicans* survival and dissemination by counteracting the barrage of antimicrobial activities associated with engulfment by phagosomes. However, instead of being favorable for *C. albicans* cells, alkalization likely hampers efficient nutrient uptake; most nutrient transport processes are driven by proton-mediated symport. Thus, fungal cells within a hostile phagosomal microenvironment, essentially devoid of high quality nutrients, likely experience increasing difficulties to obtain nutrients to support growth as the external pH increases. Based on the results presented here there is a need to clarify, and perhaps reconsider, the specific role of alkalization in *C. albicans* virulence. Clearly, there are several physiological aspects that remain unexplored and inconsistent with currently accepted dogma.

## Methodology

### Organisms, culture media, chemicals, and buffers

Strains listed in S1 Table were routinely cultivated in YPD agar medium (1% yeast extract, 2% peptone, 2% glucose, and 2% Bacto agar) at 30°C after recovery from -80°C glycerol stock. Where needed, YPD medium was supplemented with 25, 100 or 200 µg/ml nourseothricin (Nou; Jena Biosciences, Jena, Germany). Also, where indicated, the glucose in YPD is lowered to 0.2% ($YPD_{0.2\%}$) or replaced with 1% glycerol (YPG) or 2% maltose (YPM). Specific growth assays were carried out in synthetic minimal medium containing 0.17% yeast nitrogen base without ammonium sulfate and amino acids (YNB; Difco), supplemented with the indicated amino acid (10 mM) as sole nitrogen source and the indicated carbon source, and buffered to pH = 6.0 with 50 mM MES. For isotonic buffers, either PBS or HBSS both at pH = 7.4 was used. The HBSS used in this paper contains NaCl (138 mM), KCl (5.33 mM), $KH_2PO_4$ (0.44 mM), $Na_2HPO_4$ (0.3 mM), $MgCl_2$ (0.5 mM), $MgSO_4$ (0.41 mM), $CaCl_2$ (1.26 mM), $NaHCO_3$ (4 mM), HEPES (25 mM), and glucose (5.6 mM).

### Alkalization assays

Alkalization was assessed using YNB+CAA base medium composed of 0.17% YNB, 1% of casamino acids (CAA; Difco) as sole nitrogen and carbon source, and 0.01% Bromocresol Purple (BCP; Sigma) as pH indicator; the pH was set at 4.0 using 1 M HCl. Where indicated, this base medium was supplemented with ammonium sulfate (($NH_4)_2SO_4$; 38 mM) and additional carbon source such as glucose (2% or 0.2%), or glycerol (1%). For some experiments, YNB +Arg was used, which contains 10 mM arginine instead of 1% CAA. In standardized experiments, cells from overnight YPD cultures were harvested, washed at least twice in $ddH_2O$, and then suspended at an $OD_{600} \approx 0.05$ to 0.1 unless a higher starting cell density is needed. Cultures were grown with vigorous agitation at 37°C. Where appropriate, Antimycin A or Chloramphenicol was added at the concentrations indicated. In some experiments requiring phenotypic screening, purified colonies are directly inoculated from YPD agar to the indicated liquid alkalization medium and then grown 16–24 h at 37°C. For assays on solid media (with 2% agar), 5 µl aliquots of washed cell suspensions ($OD_{600} \approx 1$) were spotted onto the surface of media in a 6-well microplate and then grown at 37°C for up to 72 h. When actual pH value was needed, a 50 ml culture was used; cell pellets were first collected in a 50 ml tube and then the supernatant transferred into a fresh tube for pH measurement using a calibrated pH electrode.

### CRISPR/Cas9 mediated gene inactivation and reconstitution

CRISPR/Cas9 was used to simultaneously inactivate both alleles of *GDH2* (C2-07900W) [43,44]. Synthetic guide RNAs (sgRNAs), repair templates (RT), and verification primers used for gene editing are listed in S2 Table. Briefly, 20-bp sgRNAs primers (p1/p2), designed according to [45], were ligated to *Esp*3I (*Bsm*BI)-restricted and dephosphorylated pV1524 creating pFS108. The CRISPR/Cas9 cassette (100 ng/µl) of pFS108, with sgRNA targeting *GDH2*, was released by *Kpn*I and *Sac*I restriction and introduced into *C. albicans* together with a PCR-amplified RT (p3/p4; 100 ng/µl) containing multiple stop codons and a diagnostic *Xho*I restriction site (plasmid:repair template volume ratio of 1:3). *C. albicans* transformation was performed using the hybrid lithium acetate/DTT-electroporation method by Reuss, et al. [46]. After applying the 1.5 kV electric pulse, cells were immediately recovered in YPD medium supplemented with 1 M sorbitol for at least 4 hours, and then plated on YPD-Nou plates (200 µg/ml). Nou-resistant ($Nou^R$) transformants were re-streaked on YPD-Nou plates

(100 µg/ml) and pre-screened for the ability to alkalinize YNB+Arg media. DNA was isolated from transformants with and without alkalization defect and then subjected to PCR-restriction digest (PCR-RD) verification using primers (p5/p6) and *Xho*I restriction enzyme. The *gdh2-/-* positive clones (CFG277 and CFG278) were grown overnight in YPM to pop-out the CRISPR/ Cas9 cassette. Nou sensitive (Nou$^S$) cells were identified by plating on YPD supplemented with 25 µg/ml Nou [46], resulting in strains CFG279 and CFG281.

For reconstitution, an approximately ~1 kb gene fragment of *GDH2* with sequences homologous to the region targeted by CRISPR/Cas9 was introduced into strains CFG279 (*gdh2-/-*) and CFG354 (*cph1Δ/Δ efg1Δ/Δ gdh2-/-*). Transformants were selected on solid YNB+CAA (pH = 4.0) with BCP; the plates were incubated at 37˚C for 2–3 days. As a negative control, an unrelated gene fragment (P$_{ADH1}$-RFP-caSAT1; ≈ 5.9 kb) was independently introduced; no growth was observed on selective media. Colonies growing on YNB+CAA were re-streaked on YPD agar and retested for their ability to alkalinize the media. Genomic DNA was isolated from Alk$^+$ colonies and analyzed using PCR with primer pair (p13/p6) followed by *Xho*I restriction digest. All colonies that had regained the capacity to grow on YNB+CAA and alkalinize the media were found to be heterozygous, i.e., carrying a mutated *gdh2-* (with a *Xho*I site) and a reconstituted wildtype *GDH2* allele (lacking a *Xho*I site).

### Reporter strains

For C-terminal GFP tagging of Gdh2, an approximately 2.8 kB of PCR cassette was amplified from plasmid pFA-GFPγ-*URA3* [47] using primers (p7/p8). The amplicon was purified and then introduced into CAI4 (*ura3/ura3*) [48]. Transformants were selected on synthetic complete dextrose (CSD) plate lacking uridine. The correct integration of the GFP reporter was assessed using PCR (p5/p9), immunoblotting, and fluorescence microscopy. CFG275, a *gdh2-/-* strain constitutively expressing RFP, was constructed by introducing a *Kpn*I/*Sac*I fragment from pJA21 containing the P$_{ADH1}$-RFP-*caSAT1* construct [49] into CFG279; transformants were selected on YPD agar with 200 µg/ml Nou. Nou$^S$ recombinants lacking *CaSAT1*, the Nou$^R$ marker, were isolated after growing cells in YPM for 6 h; Nou$^S$ colonies were identified by plating cells on YPD supplemented with 25 µg/ml Nou. RFP-positive clones were verified by PCR (p11/p12) and fluorescence microscopy.

### Ammonia release assay

Quantification of volatile ammonia release was performed in accordance to the modified acid trap method by Morales et al. [50]. Briefly, a 2 µl aliquot of OD$_{600}$ ≈ 1 cell suspension was spotted onto each well of a 96-well microplate containing 150 µl of YNB+CAA solid medium supplemented with 0.2% glucose and buffered to pH = 7.4 with 50 mM MOPS. The spotted microplate was inverted and then precisely positioned on top of another microplate in which each well contains 100 µl of 10% (w/v) citric acid. Plates were sealed by parafilm and then incubated at 37˚C for 72 h after which, the citric acid solution was sampled for ammonia analysis using Nessler's reagent (Sigma-Aldrich). The solution was diluted 10-fold in ddH$_2$O and then a 20-µl aliquot was added to 80 µl Nessler's reagent on another microplate. After a 30 min incubation period at room temperature, OD$_{400}$ was measured using Enspire microplate reader. The level of ammonia entrapped in the citric acid solution was calculated based on ammonium chloride (NH$_4$Cl) standard curve.

### Filamentation assay

Filamentation in solid Spider [51] or Lee's [52] media was performed as described [53]. Cells from overnight YPD liquid cultures were harvested, washed 3X with sterile PBS, adjusted to

$OD_{600} \approx 1$ and then 5 μl of cell suspensions were spotted onto the indicated media. Plates were allowed to dry at room temperature before incubating at 37˚C as indicated.

## Subcellular localization of Gdh2

Cells expressing Gdh2-GFP (CFG273) from log-phase YPD cultures were harvested, washed 3X with ddH$_2$O, and then grown in SED$_{0.2\%}$ (10 mM glutamate and 0.2% glucose) and YNB +CAA for 24 h at 37˚C at a starting $OD_{600} \approx 0.05$. Cells from each culture were harvested, washed once with PBS, and then stained with 200 nM (in PBS) of the mitochondrial marker, MitoTracker Red (MTR; Molecular Probes) for 30 min at 37˚C. After staining, the cells were collected again and resuspended in PBS before viewing the cells using confocal microscope (LSM800, 63x oil) in the green and red channels excited with 488 nm and 561 nm lasers, respectively.

## Gdh2 expression analysis

For Gdh2 protein level analysis, cells expressing Gdh2-GFP (CFG273) were grown in liquid YPD for overnight at 30˚C and then washed 3X with ddH$_2$O. Cells were diluted in the indicated alkalization media at $OD_{600} \approx 2$ and then incubated continuously in a rotating drum for 6 h at 37˚C with sampling performed every 2 h. In each sampling point, cells were harvested, washed once with ice-cold ddH$_2$O, and then adjusted to $OD_{600} \approx 2$. Whole cell lysates were prepared using sodium hydroxide/ trichloroacetic acid (NaOH/TCA) method as described previously with minor modifications [54]. Briefly, 500 μl of adjusted cell suspension were added to tube containing 280 μl of ice-cold 2 M NaOH with 7% ß-Mercaptoethanol (ß-Me) for 15 min. Proteins were then precipitated overnight at 4˚C by adding the same volume of cold 50% TCA. Protein pellets were collected by high-speed centrifugation at 13,000 rpm for 10 min (4˚C) and then the NaOH/TCA solution completely removed. The pellets were resuspended in 50 μl of 2X SDS sample buffer with additional 5 μl of 1 M Tris Base (pH = 11) to neutralize the excess TCA. Samples were denatured at 95–100˚C for 5 min before resolving the proteins in sodium dodecyl sulfate-polyacrylamide gel electrophoresis (SDS-PAGE) using 4–12% pre-cast gels (Invitrogen). Proteins were analyzed by immunoblotting on nitrocellulose membrane according to standard procedure. After transfer, membranes were blocked using 10% skimmed milk in TBST (TBS + 0.1% Tween) for 1 h at room temperature. For Gdh2-GFP detection, membranes were first incubated with mouse anti-GFP primary antibody at 1:2,000 dilution (JL8, Takara) for overnight at 4˚C. For the detection of the primary antibody, an HRP-conjugated goat anti-mouse secondary antibody (Pierce) was used. For loading control, α-tubulin was detected with rat monoclonal antibody conjugated to HRP [YOL1/34] (Abcam). For both secondary antibody and loading control, antibodies were used at 1:10,000 dilution in 5% skimmed milk in TBST incubated for 1 h at room temperature. Immunoreactive bands were visualized by enhanced chemiluminescent detection system (SuperSignal Dura West Extended Duration Substrate; Pierce) using ChemiDoc MP system (BioRad).

To follow Gdh2 expression during alkalization in liquid culture, cells (CFG273) were prepared and collected as in immunoblotting (see preceding paragraph) but instead of cold washing following every harvesting, cells were washed once with PBS and then viewed immediately under microscope (Axio Observer 7; 20X magnification) equipped with appropriate filters to detect GFP. Where indicated, samples at specific time points were taken and analyzed to confirm subcellular localization of GFP signal using MitoTracker (see preceding section–Subcellular localization of Gdh2). For growth on solid media, cells were collected from overnight YPD culture, washed 3X with ddH$_2$O, and then adjusted to a cell density of $OD_{600} \approx 0.05$. To make a YNB+CAA agar slab on which to grow the cells, a 100 μl molten YNB+CAA agar was placed

on top of a flame-sterilized slide and then spread evenly to make a thin agar film. The agar was allowed to congeal at room temperature before spotting a 2 μl aliquot of adjusted cell suspension and then covered with a coverslip. Single cells were located immediately and then the GFP expression was followed every hour using confocal microscope (LSM800, 63x oil) in the green (GFP) channel alongside Differential Interference Contrast (DIC) for 6 h.

## Macrophage culture

Unless otherwise indicated, RAW264.7 (RAW) murine macrophage cells (ATCC TIB-71) and primary bone marrow-derived macrophages (BMDM) were cultured and passaged in complete RPMI medium supplemented with 10% fetal bovine serum (FBS), 100 U/ml penicillin and 100 mg/ml streptomycin (referred to as R10 medium in the text) in a humidified chamber set at 37°C with 5% $CO_2$. For BMDM differentiation, bone marrows collected from mouse femurs of C57BL/6 wildtype mice (7- to 9- week old) were mechanically homogenized and resuspended in R10 medium supplemented with 20% L929 conditioned media (LCM). Differentiation was carried out initially for 3 days before boosting the cells with another dose of 20% LCM until harvested. BMDM were used 7–10 days after differentiation. When needed, complete $CO_2$-independent medium (Gibco) with 10% FBS and Penn/Strep was used.

## *C. albicans* killing assay

To assess candidacidal activity by BMDM, we co-cultured *C. albicans* wildtype (PLC005) and *gdh2-/-* (CFG279) mutant with BMDM in R10 medium and then assessed colony forming units (CFU) following co-incubation. About 16–24 h prior to co-culture, differentiated BMDM were collected by scraping, counted, and then seeded at 1 x 10$^6$ cells/well into a 24-well microplate. *C. albicans* cells from overnight YPD cultures were collected by centrifugation, washed 3X with sterile PBS, and then added to macrophages at MOI 3:1 (C:M). The plates were briefly centrifuged at 500 x g for 5 min to collect the fungal cells at the bottom of each well and then co-cultured for 2 h in a humidified chamber at 37°C with 5% $CO_2$. After co-culture, each well was treated with 0.1% Triton X-100 for 2 min followed by vigorous pipetting to lyse the macrophage and release the fungal cells. Each well was rinsed seven times (7X) with ice-cold dd$H_2O$ and collected in a 15-ml conical tube. Lysates were serially diluted and then plated onto YPD. Plates containing colonies between 30–300 were counted. The candidacidal activity (% killing) of BMDM was defined as [1 - (CFU with macrophage/CFU of initial fungal inoculum] x 100 [55]. For competition assay, WT and *gdh2-/-* mutants were mixed 1:1 and then the mixed cells were used to infect BMDM at MOI of 3:1 for 2 h. After an initial macrophage lysing step, the ratio of WT: *gdh2-/-* was analyzed by plating serially diluted cells on YPD and then testing at least 50 independent colonies for their ability to alkalinize YNB+CAA medium (pH = 4). The ratio of recovered cells after 2 h was compared to the input ratio.

## Time Lapse Microscopy (TLM)

Unless otherwise indicated, all TLM experiments performed in this paper used either Cell Observer or Axio Observer 7 (both from Zeiss) inverted microscopes equipped with temperature control chambers (37°C) and $CO_2$ input. The Zeiss Cell Observer has DG4 as light source whereas the Axio Observer 7 has Colibri 5/7 as LED light source with six LED modules and seven fluorescence channels, each producing monochromatic light of a specific wavelength. For TLM using Cell Observer, the software autofocus is used to stabilize focus during image acquisition whereas Axio Observer 7 uses Definite focus.

To observe Gdh2-GFP expression during macrophage interaction, strain CFG273 was co-cultured with either RAW264.7 or BMDM macrophage pre-stained with LysoTracker Red

DND-99 (LST; Thermo Scientific) that marks the acidic compartments inside the macrophage. Macrophages were seeded at 1 x $10^6$ cells into a 35 mm glass bottom imaging dish and were allowed to adhere overnight (16–24 h). Prior to co-culture, medium was removed and then replaced with complete $CO_2$-independent medium containing 200 nM of LST. Macrophages were stained for at least 30 min at 37˚C. For fungal cell preparation, cells from overnight YPD cultures were harvested, washed 3X with sterile PBS, and then added to macrophages at MOI of 1:1 (C:M). Interaction was followed every 1.5 min for ~3 h (with RAW cells) and 2 min for ~4 h (with BMDM) using Cell Observer in the DIC, green (Gdh2-GFP) and red (LST) channels. Movies were saved at 10 fps.

To follow the behavior of WT and *gdh2-/-* during co-culture with BMDM, wildtype cells constitutively expressing GFP (SCADH1G4A) and *gdh2-/-* strain expressing RFP (CFG275) were used. Cells from overnight YPD cultures were collected by centrifugation, washed 3X with sterile PBS, and then diluted to $OD_{600} \approx 1$. Cells were either added as is to the macrophage or in the case of competition, were first mixed 1:1 (v/v) in a sterile tube followed by vigorous mixing. Prior to co-culture, BMDM (1 x $10^6$) seeded on imaging dish were washed 2X with HBSS to remove the growth medium. A 100 μl aliquot of single or mixed fungal cells (~3 x $10^6$ cells) were added to the dish (MOI of 3:1, C:M) and phagocytosis was carried out in HBSS for approximately 30 min in the humidified chamber. Co-cultures were washed at least 5X with HBSS and 1X with $CO_2$-independent medium to remove non-phagocytized fungal cells. $CO_2$-independent medium was added to the dish and TLM was carried out at 37˚C using Cell Observer (competition) or Axio Observer 7 (individual strain) in the DIC, green (wildtype) and red (*gdh2-/-*) channels. Images were acquired every 2 min for at least 5 h.

We note that time-lapse microscopy has a distinct advantage over endpoint microscopy of fixed co-cultures, since it enables the spatio-temporal dynamics of hyphal formation of both wildtype and *gdh2-/-* mutant to be accurately followed inside the macrophage. Time-lapse microscopy allowed us to observe that non-phagocytized fungal cells that remained external even after excessive washing can filament resulting in the false impression that the fungal cells are escaping from the macrophage.

## Monitoring phagosomal pH

To monitor the phagosomal pH in macrophages after engulfment of *C. albicans* cells, we used the pH-sensitive pHrodo Red, succinimidyl ester (pHrodo Red, SE) from Life Technologies (Oregon, USA). The fluorescence intensity of pHrodo-labeled cells increases when pH decreases. Mid log phase cells in YPD were harvested, washed 3X with PBS, and then adjusted to $OD_{600} = 10$. Cells were collected from 1 ml of adjusted cell suspension and then resuspended thoroughly in 800 μl of 100 mM $NaHCO_3$ buffer (pH = 8.5). A 400 μl aliquot was stained with 125 μM of pHrodo for 1 h at room temp with gentle shaking protected from light. DMSO was used for unstained control. After incubation, cells were carefully washed with PBS at least four times to remove the excess dye, and then resuspended in 400 μl of PBS. Cells were then opsonized with 1:400 dilution of purified anti-*Candida albicans* antibody (OriGene, catalogue no. BP1006) for 30 min at 37˚C prior to infection at the MOI indicated in the text. To assess the effect of pHrodo on viability, labeled cells were serially diluted and spotted on a YPD plate and then grown at 30˚C for 2–3 days. For emission scans, a 10 μl aliquot of labeled cells was diluted in 190 μl of the indicated buffer and analyzed at 37˚C using Enspire multimode plate reader (Perkin Elmer) with excitation of 532 nm.

For co-culture, RAW macrophages were cultured in complete DMEM medium (high glucose, no phenol red, with HEPES and bicarbonate; Gibco) supplemented with 10% FBS and Penn/Strep. RAW cells were collected and then seeded at 1 x $10^6$ cells in an ibidi imaging dish

(μ-Dish 35 mm high; ibidi, GmbH Germany) with DIC-compatible cover and were allowed to adhere on the dish overnight in a humidified $CO_2$-incubator at 37˚C. Prior to infection, RAW cells were washed once with prewarmed HBSS and then replaced with 2.5 ml of fresh HBSS. Dishes were immediately placed in the sample port of the Zeiss Axio Observer 7 microscope, equipped with an OKOlab incubation chamber preheated to 37˚C and conditioned at 5% $CO_2$. The 555 nm LED of the Colibri 7 LED light source was used for pHrodo excitation, and emission light was captured using a 575 nm beam splitter and appropriate filter sets to collect emission signal at 585 nm (DFT 490/575). Images were acquired using a Hamamatsu Orca fusion prime BSI camera and Zeiss Definite Focus hardware for focus stabilization during TLM. pHrodo-labeled cells were immediately added to the dish and images captured either every 30 min or every 2 min for time lapse movies at 20X magnification. Phagosome region of interests (ROIs) were identified initially by masking and then ROIs manually verified in the DIC channel. For intensity quantification involving time lapse movie the Time Series Analyzer V3 plug-in in Fiji software was used. For calibration of pHrodo intensity against pH, 2 μl of stained cells were added to 2.5 ml of each specific buffer: sodium citrate buffer (pH = 4 and 5), MES (pH = 6), MOPS (pH = 7), and HEPES (pH = 8), equilibrated for 10 min to collect the cells at the bottom of the dish, and then imaged without $CO_2$. pHrodo intensities of either individual fungal cells or phagosomes were quantified from uncompressed raw TIFF data using Fiji software. For display of pHrodo intensities, a rainbow LUT in the Zen software was used.

## Murine systemic infection model

Groups of female C57BL/6 mice, aged 6–8 weeks, were purchased from Beijing Vital River Laboratory Animal Technology Co., Ltd. These mice were housed in individual ventilated cages in a pathogen-free animal facility at Institut Pasteur of Shanghai, Chinese Academy of Sciences. In each of the mouse studies, the animals were assigned to the different experimental groups. Infections were performed under SPF conditions. Wildtype SC5314 and *gdh2-/-* mutant *C. albicans* cells were inoculated into YPD broth and grown overnight at 30˚C. Cells were harvested and washed three times with phosphate-buffered saline (PBS), and counted using hemocytometer. For each strain, mice (n = 10) were injected via the lateral tail vein with $3x10^5$ CFU of *C. albicans* cells. The mice were monitored once daily for weight loss, disease severity and survival. The fungal burden was assessed by counting CFU. The survival curves were statistically analyzed by the Kaplan-Meier method (a log-rank test, GraphPad Prism). Competitive bloodstream infections were performed using equal numbers of SC5314 and *gdh2-/-* mutant cells, i.e., with an inoculum (I) ratio of 1:1. At 3 days post infection the abundance and genotype of fungal cells recovered from kidneys was determined and the ratio of wildtype:*gdh2-/-* in kidneys was calculated (R). Cells lacking *gdh2-/-* cannot grow on selective YNB+Arg medium. The $\log_2(R/I)$ values was compared using unpaired t-test.

## Ethics statement

All animal experiments were carried out in strict accordance with the Regulations for the Administration of Affairs Concerning Experimental Animals issued by the Ministry of Science and Technology of the People's Republic of China, and approved by IACUC at the Institut Pasteur of Shanghai, Chinese Academy of Science with an approval number P2018050.

## Supporting information

**S1 Fig. CRISPR/Cas9-mediated gene inactivation of *GDH2* in *C. albicans*.** (A) A purified *Kpn*I/*Sac*I fragment from pFS108, harboring *GDH2*-specific sgRNA, and PCR generated repair template (RT) were introduced into wildtype strain SC5314 by electroporation. Nou[R]

transformants were pre-screened in YNB+Arg medium containing the pH indicator bromo-cresol purple; the initial pH was 4.0. Three Nou$^R$ colonies were picked for further analysis. Clones #1 and #2 grew poorly and were unable to alkalinize the media; clone #3 grew and alkalinized the media. (B) Genomic DNA, isolated from the three clones, was used as template for PCR amplification of the targeted *GDH2* locus; ddH$_2$O was used as negative control. Restriction of the amplified ≈900 bp fragment by *Xho*I is diagnostic for successful mutagenesis. Strains, clone #1 (CFG277) and clone #2 (CFG278) carry inactivated *gdh2-/-* alleles. (C) *GDH2* is not essential but required for robust growth on glutamate or proline as sole nitrogen source. Five microliters of serially diluted wildtype (SC5314), *gdh2-/-* NAT$^R$ (CFG277), *gdh2-/-* NAT$^S$ (CFG279), and control (CFG182) cells were spotted on yeast peptone (YP), synthetic gluta-mate (SE) and synthetic proline (SP) media containing either 2% glucose (D) or 1% glycerol (G) as carbon source. The plates were incubated for 48 h at 30°C and photographed. (D) A colony from fresh YPD plates were directly inoculated onto a single well of a 96-well plate containing YNB+CAA medium and then incubated for 24 h at 37°C.
(TIF)

**S2 Fig. Amino acid-dependent alkalization is abolished in *gdh2-/-* mutant.** Cells of the indicated strains were collected from YPD overnight culture, washed, and then diluted to OD$_{600}$ ≈ 5.0 in YNB+CAA medium supplemented with 38 mM ammonium sulfate (AS) and 1% glycerol. Tubes were incubated at 37°C and then photographed at the indicated times. Only *gdh2-/-* mutant failed to alkalinize the medium. Strains used: WT (SC5314), *put1-/-* (CFG154), *put2-/-* (CFG318), *gdh2-/-* (CFG279) and *stp2Δ/Δ* (SVC17).
(TIF)

**S3 Fig. *GDH2* reconstitution in *gdh2-/-* strains.** (A) Strains CFG279 (*gdh2-/-*) and CFG354 (*cph1Δ/Δ efg1Δ/Δ gdh2-/-*) were transformed with wildtype *GDH2* gene fragment that encompasses the mutated region of *GDH2*. An unrelated gene fragment (pJA21, P$_{ADH1}$-*RFP-caSAT1*) was also used to transform the same strains as control. Transformants were selected on YNB +CAA with BCP and 2% agar (pH = 4.0) as control. Transformation plate images taken 2-3 days after incubation at 37°C showing alkalization positive transformants. (B) Purified colonies from (A) were verified by PCR followed by *Xho*I restriction digest (RD). Heterozygotes (*gdh2-/ GDH2*) were identified by the presence of both mutated and wildtype *GDH2* alleles. PCR-RD verification of *GDH2/gdh2-* (Top) and *cph1Δ/Δ efg1Δ/Δ GDH2/gdh2-* (Bottom) reconstituted strains. Clones S2, S4, S5, and S19 (Top) and clones 1 and 2 (Bottom) were obtained from a separate transformation experiment were following electroporation, cells were directly recovered and enriched in liquid YNB+CAA for 24 h at 37°C prior to plating on YPD agar for single colonies. (C) All strains were verified again in their capacity to grow and alkalinize the YNB+CAA medium by directly inoculating purified colonies into each well containing medium and then grown statically at 37°C for 24 h. Reconstituted strains shown were randomly selected from the PCR-RD positive clones. Wells in the SC5314 lane: **1** (PLC005), **2** (CFG279), **3** (CFG355; Clone S5), **4** (CFG356; Clone S19), **5** (CFG357; Clone T2s1), **6** (CFG358; Clone T2-2), **7** (*gdh2-/ GDH2*; Clone T2s2), **8** (*GDH2/gdh2-*; Clone A), and **9** (Medium); wells in the *cph1Δ/Δ efg1Δ/Δ* lane: **1** (CASJ041), **2** (CFG354), **3** (CFG359; Clone 1), **4** (CFG360; Clone 2), **5** (CFG361; Clone 3), **6** (CFG362; Clone 4), **7** (*cph1Δ/Δ efg1Δ/Δ GDH2/gdh2-*; Clone 6), **8** (*cph1Δ/Δ efg1Δ/Δ GDH2/gdh2-*; Clone 8), and **9** (Medium). Reconstituted strains containing the "CFG-" code were stored at -80°C as glycerol stocks and were listed in the strains list.
(TIF)

**S4 Fig. Mitochondrial inhibition by Antimycin A inhibits alkalization in wildtype cells.** Wildtype cells (SC5314) collected from YPD overnight cultures were washed and then diluted

to $OD_{600} \approx 5$ in liquid YNB+CAA medium supplemented with 38 mM ammonium sulfate (AS) and 1% glycerol with the indicated concentrations of mitochondrial complex III inhibitor antimycin A. Cultures were incubated continuously at 37˚C under constant aeration and then photographed at the indicated time points. Images are representative of at least 3 independent experiments. For control (-), equal amount of ethanol carrier as that of 1.5 μg/ml of antimycin A was added to the tube.
(TIF)

**S5 Fig. Growth characteristics of wildtype and *gdh2-/-* strains in liquid YNB+CAA with and without glucose and chloramphenicol (Cm).** (A) Gdh2-dependent alkalization is sensitive to glucose (Left panel). YPD grown wildtype (WT, SC5314) and *gdh2-/-* (CFG279) cells were collected, washed, and diluted to an $OD_{600} \approx 0.05$ in YNB+CAA with 0, 2 or 0.2% glucose as indicated. The cultures were grown under vigorous agitation at 37˚C for 16 h and the pH was measured (the initial pH was 4.0; the values indicated are the average of three replicate cultures). Alkalization is linked to mitochondrial function (Right panel). Wildtype cells (SC5314) from overnight YPD cultures were washed and diluted to $OD_{600} \approx 0.1$ in liquid YNB+CAA (0.2% glucose) with the indicated concentrations of mitochondrial translation inhibitor chloramphenicol. Cultures were grown at 37˚C under vigorous agitation for 16 h. (B) Phenotypic validation of the reporter strains used in macrophage co-cultures. Growth of wildtype (WT; $P_{ADH1}$-*GFP-caSAT1*; SCADH1G4A) and *gdh2-/-* ($P_{ADH1}$-*RFP-caSAT1*, CFG275) cells in YNB +CAA supplemented with 0.2% glucose. Cultures were grown for 16 h at 37˚C.
(TIF)

**S6 Fig. Gdh2 expression is sensitive to glucose.** (A) Cells expressing Gdh2-GFP (CFG273) were collected from YPD overnight cultures, washed, and then diluted in liquid YNB+CAA with or without the indicated concentrations of glucose or glycerol at $OD_{600} \approx 2.0$ and then incubated under aeration at 37˚C. Cells were harvested at the indicated time points and then immediately washed with $ddH_2O$ for microscopic examination of Gdh2-GFP expression. Representative images of cells from each condition with their relative expression of Gdh2-GFP are shown. (Bottom panel) Cells collected from one of the conditions (i.e., no addition; T = 4h) were stained with MitoTracker Deep Red (MTR; 200 nM) and then observed by confocal microscopy (LSM800) using the Airyscan detector. The Gdh2-GFP signal colocalizes with MTR. (B) Quantification of mean fluorescence intensity (MFI) from 3 biological replicates per condition ($\geq$150 cells/replicate) are shown.
(TIF)

**S7 Fig. pH dependence of pHrodo intensity.** Mid log phase cells from YPD culture were harvested, washed, and then stained with pHrodo (or DMSO as control) in $NaHCO_3$ buffer as outlined (Methodology). (A) Viability assessment of cells stained with the dye and then opsonized prior to infection. Stained cells were serially diluted in PBS and then an aliquot (5 μl) spotted on YPD. Photographs taken after 48 h of growth at 30˚C. (B) A 2-μl aliquot of stained cells were added to imaging dish containing 2.5 ml of the buffer, equilibrated for at least 5 min, and then cells were imaged at 37˚C. Quantification of mean fluorescence intensity (MFI) from 3 biological replicates per condition ($\geq$150 cells/replicate) are shown. (Ave. ± CI; **** p £ 0.0001 by one-way ANOVA). (C) Ten-μl aliquots of samples were added to 190 μl of buffer and analyzed for emission using Enspire reader with excitation of 532 nm.
(TIF)

**S8 Fig. Confirmation of acidified phagosomes by ionophore treatment.** (Top) pHrodo-stained wildtype (PLC005) and *gdh2-/-* (CFG279) cells were co-cultured with macrophage for around 4 h and then treated with 10 μM of both monensin and nigericin (i.e., Ionophores,

Ion) for 5 min to dissipate proton gradients in acidified compartments. Co-cultures were photographed before and after addition of ionophores. (Bottom) Quantification of phagosome intensities before and after addition of ionophores. At least 100 phagosomes/replicate were analyzed (Ave. ± CI; **** p £ 0.0001 by Student *t*-test).
(TIF)

**S1 Table. Strains used in this study.**
(DOCX)

**S2 Table. Primers used in this study.**
(DOCX)

**S1 Vid. Gdh2-GFP (CFG273) is induced upon phagocytosis by RAW264.7 macrophages (Fig 4B).**
(WMV)

**S2 Vid. Gdh2-GFP (CFG273) is induced upon phagocytosis by primary murine BMDM.**
(WMV)

**S3 Vid. Filamentation of wildtype (SCADH1G4A) upon phagocytosis by BMDM (Fig 5A).**
(M4V)

**S4 Vid. Filamentation of *gdh2*-/- (CFG275) upon phagocytosis by BMDM (Fig 5B).**
(M4V)

**S5 Vid. Competition assay to compare wildtype (SCADH1G4A) and *gdh2*-/- (CFG275) filamentation and survival upon phagocytosis by BMDM (Fig 5B).**
(ZIP)

**S6 Vid. Time-lapse microscopy (TLM) movie of pHrodo-stained wildtype cells (PLC005) interacting with RAW264.7 macrophage performed in HBSS (pH = 7.4) at 37˚C and 5% $CO_2$.** Images were taken every 2 min for ~4 h. MOI of 3:1 (C:M) was used.
(M4V)

**S7 Vid. Time-lapse microscopy (TLM) movie of pHrodo-stained *gdh2*-/- cells (CFG279) interacting with RAW264.7 macrophage performed in HBSS (pH = 7.4) at 37˚C and 5% $CO_2$.** Images were taken every 2 min for ~4 h. MOI of 3:1 (C:M) was used.
(M4V)

**S8 Vid. Hyphal induction in acidified phagosomes containing WT cells (PLC005).**
(AVI)

**S9 Vid. Hyphal induction in acidified phagosomes containing *gdh2*-/- cells (CFG279).**
(AVI)

**S10 Vid. Time-lapse microscopy (TLM) of pHrodo-stained wildtype (PLC005) co-cultured with RAW macrophage at high MOI (6:1; C:M).**
(M4V)

**S11 Vid. Time-lapse microscopy (TLM) of pHrodo-stained *gdh2*-/- (CFG279) co-cultured with RAW macrophage at high MOI (6:1; C:M).**
(M4V)

**S12 Vid. Monitoring of pHrodo intensity changes in a single phagocytic event (*gdh2*-/-).**
(AVI)

## Acknowledgments

The authors would like to thank the members of the Claes Andréasson, Sabrina Büttner, Roger Karlsson and Per Ljungdahl laboratories (SU) for their constructive comments throughout the course of this work. Gratitude is extended to Valmik Vyas and Gerard Fink (MIT, Cambridge, MA, USA) for providing the CRISPR/Cas9 cassettes, Joachim Morschhäuser (Universität Würzburg, Germany), and Slavena Vylkova (Hans Knöll Institute, Jena, Germany) for supplying strains and for fruitful discussions. We also thank Stina Höglund, the Imaging Facility-Stockholm University (IFSU), for assistance in microscopy. We would also like to acknowledge Andreas Ring (SU), Joachim Morschhäuser (Universität Würzburg) and Johannes Westman (Hospital for Sick Children, Toronto, ON, CA) for constructive comments on the manuscript.

## Author Contributions

**Conceptualization:** Fitz Gerald S. Silao, Meliza Ward, Chris Molenaar, Ning-Ning Liu, Changbin Chen, Per O. Ljungdahl.

**Data curation:** Fitz Gerald S. Silao, Kicki Ryman, Tong Jiang, Meliza Ward, Nicolas Hansmann, Chris Molenaar, Ning-Ning Liu.

**Formal analysis:** Fitz Gerald S. Silao, Kicki Ryman, Tong Jiang, Meliza Ward, Chris Molenaar, Ning-Ning Liu, Changbin Chen, Per O. Ljungdahl.

**Funding acquisition:** Per O. Ljungdahl.

**Investigation:** Fitz Gerald S. Silao, Kicki Ryman, Tong Jiang, Meliza Ward, Nicolas Hansmann, Chris Molenaar.

**Methodology:** Fitz Gerald S. Silao, Chris Molenaar.

**Project administration:** Per O. Ljungdahl.

**Resources:** Per O. Ljungdahl.

**Supervision:** Fitz Gerald S. Silao, Ning-Ning Liu, Changbin Chen, Per O. Ljungdahl.

**Visualization:** Fitz Gerald S. Silao, Kicki Ryman, Tong Jiang, Nicolas Hansmann, Chris Molenaar, Ning-Ning Liu, Changbin Chen, Per O. Ljungdahl.

**Writing – original draft:** Fitz Gerald S. Silao.

**Writing – review & editing:** Fitz Gerald S. Silao, Chris Molenaar, Per O. Ljungdahl.

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
