## [Editor Report · Decision Letter 0]

7 Feb 2020

Dear Professor Ljungdahl,

Thank you very much for submitting your manuscript "Glutamate dehydrogenase (Gdh2)-dependent alkalization is dispensable for escape from macrophages and virulence of Candida albicans" (PPATHOGENS-D-20-00074) for consideration at PLOS Pathogens. As with all papers, your manuscript was reviewed by members of the editorial board. Based on our initial assessment, we regret that we will not be pursuing this manuscript for publication at PLOS Pathogens. 

The consensus was that a significant limitation of the work is that the mutants were not shown to be alkalinization-defective after engulfment by macrophages. Without this critical control, it cannot be concluded definitively that alkalinization of the phagosome is dispensable for escape from macrophages and virulence of Candida albicans. 

If you are able to address this issue, we would be happy to reconsider a revised version of the manuscript.

We are sorry that we cannot be more positive on this occasion. We very much appreciate your wish to present your work in one of PLOS's Open Access publications. Thank you for your support, and we hope that you will consider PLOS Pathogens for other submissions in the future.

Sincerely,

Scott Filler

Section Editor

PLOS Pathogens

Kasturi Haldar

Editor-in-Chief

PLOS Pathogens

orcid.org/0000-0001-5065-158X

Michael Malim

Editor-in-Chief

PLOS Pathogens

orcid.org/0000-0002-7699-2064

---

## [Decision Letter · Decision Letter 1]

2 Apr 2020

Dear Dr. Ljungdahl,

Thank you very much for submitting your manuscript "Glutamate dehydrogenase (Gdh2)-dependent alkalization is dispensable for escape from macrophages and virulence of Candida albicans" for consideration at PLOS Pathogens. As with all papers reviewed by the journal, your manuscript was reviewed by members of the editorial board and by several independent reviewers. In light of the reviews (below this email), we would like to invite the resubmission of a significantly-revised version that takes into account the reviewers' comments.

All three reviewer's indicated that direct measurement of the pH within the phagolysosome should be performed and I agree with this assessment.  If you decide to do this experiment, I think an important control is strains previously reported by the group with an alternative model.  This will insure that there are not some method or strain based idiosyncrasies responsible for the discordant results. I also feel that, in interests of collegiality, a discussion of possible reasons for the discordant conclusions would be helpful  As you will see there are a number of other important controls and data that need to be reported and I encourage you to give these points attention if you decide to revise the manuscript.

We cannot make any decision about publication until we have seen the revised manuscript and your response to the reviewers' comments. Your revised manuscript is also likely to be sent to reviewers for further evaluation.

Sincerely,

Damian J Krysan, MD PhD

Associate Editor

PLOS Pathogens

Scott Filler

Section Editor

PLOS Pathogens

Kasturi Haldar

Editor-in-Chief

PLOS Pathogens

orcid.org/0000-0001-5065-158X

Michael Malim

Editor-in-Chief

PLOS Pathogens

orcid.org/0000-0002-7699-2064

Reviewer's Responses to Questions

**Part I - Summary**

Reviewer #1: This manuscript describes a role for the C. albicans Glutamate dehydrogenase (Gdh2) in alkalization of the external environment, and surprisingly shows that this function is dispensable for escape from macrophages and virulence. This was unexpected because a previous model by Lorenz et al proposed that growth on amino acids in the phagosome would lead to ammonia release and alkalinization, which would stimulate hyphal growth and escape from the phagosome. Therefore, it was surprising that that deletion of GDH2, which prevented alkalinization in vitro, did not affect escape from macrophages or virulence. One strength of the manuscript is that the data provide strong support for a role for Gdh2 in alkalinization for cells grown on amino acids in vitro. This work also synergizes with previous work by this group of investigators showing that proline catabolism, which goes through Gdh2 and glutamate, is important for growth on amino acids and alkalinization. Another interesting conclusion was the demonstration that glucose could repress mitochondrial function, and GDH2 expression, preventing alkalization of the environment when amino acids were present. However, a weakness is that the pH of the phagosome was not measured. It is not clear if metabolism of other metabolites by gdh2 mutant cells still leads to alkalinization of the phagosome. For example, GlcNAc, alpha-ketoglutarate, and pyruvate have also been observed to contribute to alkalinization.

Reviewer #2: This manuscript describes an investigation of the factors required by C. albicans for alkalization of its environment. Alkalization of the phagosome environment upon engulfment of C. albicans has previously been hypothesised to induce the yeast to hyphal switch facilitating C. albicans escape from immune cells and therefore of high importance to pathogenicity.

Here, in agreement with a previous study (Westman et al, 2018), the authors show that alkalization is not required for hyphal induction inside the phagosome. The authors demonstrate that a mutant strain lacking Gdh2 glutamate dehydrogenase is defective in alkalization of a synthetic amino acid containing medium in contrast to wild type cells. GDh2-GFP was localised to mitochondria and expression was repressed in high concentrations of glucose but induced upon macrophage engulfment. The alkalization defect did not result in a morphogenesis defect in vitro and in macrophage co-culture assays. The ghd2 mutant was also virulent in a mouse and Drosophila infection models. In S. cerevisiae Ghd2 is also glucose –repressed and mitochondrial.

Reviewer #3: This manuscript characterized the role Candida albicans Gdh2 in alkalization under conditions of amino acids as a sole carbon source. The finding of a requirement for production of NH3 under these conditions in interesting, particularly as disruption has no impact on phagocytosis or virulence.

Strengths:

Inclusion of the GFP-strain to track Gdh2 to mitochondria

The inclusion of two animal models and multiple experiments to measure virulence

Most experiment use multiple lines of evidence

Limitations:

It is unclear gdh2 is responsible for NH3 production in the phagosome

It is unclear if disruption of gdh2 influences the phagosome pH.

**Part II – Major Issues: Key Experiments Required for Acceptance**

Reviewer #1: 1. The gdh2 mutant strain was not complemented by reintroduction of the wild type GDH2 gene. This control is especially important, considering that the authors come to a different conclusion than previous studies.

2. As mentioned above, the pH of the phagosome was not measured, so it is unclear if there are alternative pathways that can lead to alkalinization in the absence of GDH2.

3. Gdh2-GFP looks different in Figure 3A, than 2A. It looks cytoplasmic in Fig. 3A rather than mitochondrial. This raises questions about whether Fig. 3A is accurately reflecting the levels and localization of Gdh2.

Reviewer #2: A key experiment as set out in the authors’ covering letter is to show that the ghd2 mutant is incapable of alkalizing the macrophage phagosome. This will strengthen the evidence presented and clearly confirm the role of Ghd2 in this process in vivo ; reinforcing the in vitro measurements of pH changes presented in the manuscript.

Reviewer #3: Figure: 1C and 1D: For figure 1D, it appears that YNB +CAA +0.2% glucose was used from the text and should be included in the legend. It is not clear how well the gdh2 mutant grow in this media. If it is not growing well, this may impact NH3 production. A growth control should be used.

Line 302: Can the authors add data showing that the phagosome is or is not alkalinized when the gdh2 mutant is engulfed?

**Part III – Minor Issues: Editorial and Data Presentation Modifications**

Reviewer #1: 1. The text should be modified to indicate that the lack of an essential role for in macrophage escape is not the same as saying it can’t contribute to escape along with other pathways.

2. In Figure 6, was the CFU assay done with macrophages infected with a single strain or mixed culture? If a WT and a gdh2-/- cell are in the same macrophage, the WT cell could alkalinize the phagosome and protect the gdh2-/- mutant.

3. It is not obvious that Gdh2-GFP is getting brighter in Figures SV1 SV2 and Figure 5. The level of Gdh2-GFP should be quantified to support the conclusion Gdh2 is induced.

4. The authors should comment on why there was only modest effect on growth of gdh2-/- mutant on proline in Fig. S1C? If Gdh2 is only partially necessary for growth on proline, does this indicate a gdh2-/- mutant could still contribute to alkalinization in a phagosome, albeit more weakly?

5. Does it matter which medium the C. albicans cells were grown in before they were used to infect macrophages? Residual carbon, nitrogen, and energy sources may dominate in the first stages of growth in the macrophage. This could make it more difficult to assess the role of Gdh2.

6. How was FITC staining done? Please provide additional information. E.g. was it crosslinked onto the cells? Taken up by endocytosis? The reason I am asking is that Figure 4B looks weird. The entire cell is stained. I was expecting new growth should not be stained if the FITC is cross-linked

7. Lines 172-174. This sentence should be clarified that Gdh2 is primarily, but not solely, responsible for ammonia production when cells were grown on glutamate.

Reviewer #2: Supplementary video 3 very rapidly goes out of focus so is not informative.

Line 192 strain

Line 200. Colonies were grown on solid……and the levels of volatile ammonia produced were measured.

Line 218 medium

Line 269 define NCR

Line 310 macrophages

Phagocytized and phagocytosed (line 325) both used in text, keep consistent

Reviewer #3: Figure S1: Why was the screen performed? Why were the mutants with the correct insertion not initially identified by PCR?

The authors should expand on how the growth media and conditions in the current study do or do not represent phagosome conditions.

Figure 2A: Include a glucose-rich condition.

Figure 2B: What is the impact of actimycin alone on the media? I assume that these experiments use the same pH indicator as 1B, but this should be added to the legend.

Lines 226-234: The description of the figure does not appear to match Figure 2. I don’t see right and left panels or the gdh2 mutant in Figure 2B.

Figure S2A: For the right panel, clarify the difference between Cm – and Cm 0. I assume that no organism was added to the Cm-, but this should be clarified.

Line 245: The authors should describe why different results were found for actimycin and chloramphenicol

Line 270: Clarify NCR

Figure 5A and B and supplemental video: Is there a peak fluorescence that is reached prior to the end of the experiment? If so, can the authors speculate as to why this is? Are the phagocytes dying and the glucose repression ending?

Line 337: Clarify the statement, as alkalization was not measured in the phagosome.

Line 378: The authors should avoid discussing the unpublished data, or include the data.

Line 448: Clarify if these conditions for filamentation were the same as the ones used in the current investigation and consider having a supplemental figure with the strains.

PLOS authors have the option to publish the peer review history of their article (what does this mean?). If published, this will include your full peer review and any attached files.

Reviewer #1: No

Reviewer #2: No

Reviewer #3: No
---

## [Decision Letter · Decision Letter 2]

13 Jul 2020

Dear Dr. Ljungdahl,

Thank you very much for submitting your manuscript "Glutamate dehydrogenase (Gdh2)-dependent alkalization is dispensable for escape from macrophages and virulence of Candida albicans" for consideration at PLOS Pathogens. As with all papers reviewed by the journal, your manuscript was reviewed by members of the editorial board and by several independent reviewers. The reviewers appreciated the attention to an important topic. Based on the reviews, we are likely to accept this manuscript for publication, providing that you modify the manuscript according to the review recommendations.

Thank you for your careful and response to previous reviews. The new data on intracellular/macrophage pH have strengthened the manuscript and both reviewers are very positive. As you will note, Reviewer 1 has a couple of minor textural suggestions that I also think should be addressed. I think this is an important and complex problem that as, you note in the discussion, has a number of interesting and outstanding questions. I appreciated your discussion of potential reasons for conflicting data and models. In the interest of collegiality and progress in the field this is very important.

Sincerely,

Damian J Krysan, MD PhD

Associate Editor

PLOS Pathogens

Scott Filler

Section Editor

PLOS Pathogens

Kasturi Haldar

Editor-in-Chief

PLOS Pathogens

orcid.org/0000-0001-5065-158X

Michael Malim

Editor-in-Chief

PLOS Pathogens

orcid.org/0000-0002-7699-2064

Reviewer Comments (if any, and for reference):

Reviewer's Responses to Questions

**Part I - Summary**

Reviewer #1: A major improvement was the inclusion of new data demonstrating the acidification of the lysosomes following C. albicans phagocytosis. The authors used a new approach of crosslinking a pH sensitive dye to the cells (pHrodo). A strength of this approach is that the fluorescence increases as the ambient pH gets lower. This avoids concerns that factors such as photobleaching could interfere.

However, there are some minor issues with the manuscript that need to be addressed. In regard to the comments below concerning Gdh2-GFP, the authors should be clearer about what are the limits of their conclusions regarding the levels of Gdh2-GFP. Also, they should explain why in some experiments Gdh2-GFP does not appear to localize to the mitochondria.

Reviewer #3: The authors have preformed the additional requested studies and addressed the comments of the reviewers.

**Part II – Major Issues: Key Experiments Required for Acceptance**

Reviewer #1: (No Response)

Reviewer #3: (No Response)

**Part III – Minor Issues: Editorial and Data Presentation Modifications**

Reviewer #1: 1. Lines 380-381,393, 514 and 1048. The manuscript refers to a supplemental video SV12, but SV12 was not included.

2. The authors complemented the gdh2-/- strain by reintroducing a DNA fragment carrying the wild type GDH2 gene. Was it marked in any way to distinguish it from a pre-existing gdh2-/GDH2 strain?

3. Fig. 3A. The Gdh2-GFP signal goes up, but it does not look mitochondrial

4. Fig S6. The Gdh2-GFP signal goes up, but it does not look mitochondrial

5. Fig. 4B and 4C. Not clear that the Gdh2-GFP signal is increasing. How was this quantified.

6. SV2 Not clear that the Gdh2-GFP is increasing until the hyphae break out of the macrophage. How was this quantified?

Reviewer #3: (No Response)

PLOS authors have the option to publish the peer review history of their article (what does this mean?). If published, this will include your full peer review and any attached files.

Reviewer #1: No

Reviewer #3: No
---

## [Editor Report · Decision Letter 3]

14 Aug 2020

Dear Dr. Ljungdahl,

We are pleased to inform you that your manuscript 'Glutamate dehydrogenase (Gdh2)-dependent alkalization is dispensable for escape from macrophages and virulence of Candida albicans' has been provisionally accepted for publication in PLOS Pathogens. Thank you for your response to the reviews and congratulations on a nice piece of work.

Best regards,

Damian J Krysan, MD PhD

Associate Editor

PLOS Pathogens

Scott Filler

Section Editor

PLOS Pathogens

Kasturi Haldar

Editor-in-Chief

PLOS Pathogens

orcid.org/0000-0001-5065-158X

Michael Malim

Editor-in-Chief

PLOS Pathogens

orcid.org/0000-0002-7699-2064
---

## [Editor Report · Acceptance letter]

4 Sep 2020

Dear Professor Ljungdahl,

We are delighted to inform you that your manuscript, "Glutamate dehydrogenase (Gdh2)-dependent alkalization is dispensable for escape from macrophages and virulence of *Candida albicans*," has been formally accepted for publication in PLOS Pathogens.

Best regards,

Kasturi Haldar

Editor-in-Chief

PLOS Pathogens

orcid.org/0000-0001-5065-158X

Michael Malim

Editor-in-Chief

PLOS Pathogens

orcid.org/0000-0002-7699-2064